# Entomopathogenic Fungi in the Soils of China and Their Bioactivity against Striped Flea Beetles *Phyllotreta striolata*

Ke Zhang , Xiaofeng Zhang, Qiongbo Hu * and Qunfang Weng *

Key Laboratory of Bio-Pesticide Innovation and Application of Guangdong Province, College of Plant Protection, South China Agricultural University, Guangzhou 510642, China; zhangke@scau.edu.cn (K.Z.); zhangxiaof0608@163.com (X.Z.)

* Correspondence: hqbscau@scau.edu.cn (Q.H.); wengweng@scau.edu.cn (Q.W.)

**Abstract:** The present research aims to explore the occurrence and diversity of entomopathogenic fungi (EPF) in cultivated and uncultivated lands from different provinces of China and to search for EPF against *Phyllotreta striolata*. In this study, first, the EPF biodiversity from the soil of four provinces (Hunan, Hubei, Henan and Hebei) was surveyed. There were 302 fungal isolates obtained from 226 soil samples collected from croplands (114), arbor (79), grasslands (97) and fallow land (12); 188 EPF isolates were identified as 11 genera. The data indicate that Hubei Province has the greatest EPF diversity, with a Shannon Evenness Index (SHEI) value of 0.88. Here, the grassland, arbor and cropland had an EPF diversity with SHEI values of 0.81, 0.86 and 0.76, respectively, while the fallow land had the highest SHEI value of 1.00, which suggests that cultivation by humans affected the count and richness of soil fungi: the less human activity, the more kinds of fungi found. Finally, the pathogenicity of 47 fungal strains against the adult *P. striolata* was determined. *Isaria javanica* (IsjaHN3002) had the highest mortality. In conclusion, this study reports the EPF distribution and biodiversity in the soil from four provinces in China, showing that the amount and type of fungi in the soil varied by region and vegetation and that soil was one of the resources for acquiring EPF. The potential of *I. javanica* as a biocontrol must be studied further.

**Keywords:** *Isaria javanica*; pathogenicity

## 1. Introduction

Entomopathogenic fungi (EPFs) are ubiquitous in nature. Biological plant protection with EPFs plays a key role in sustainable pest management programs [1]. In addition to absorbing nutrients for their own growth, some EPFs can control insect populations at low levels for long periods [2]. Fungi-based insecticides have great potential as a form of pest control [3]. Not only are EPFs are harmless to human beings, animals and crops, but they also have the advantages of long-term validity, non-resistance, no residue, no pollution, no damage to natural enemies, high epidemic potential and ease of production [4,5]. Therefore, using EPFs to control agricultural and forestry pests has become a new trend in pest control. EPFs are the largest group of insect-pathogenic microorganisms. According to incomplete statistics, about 100 genera and 1000 species of EPFs have been recorded around the world [6], and more than 40 genera and more than 400 species have been found in China [7], including *Beauveria*, *Metarhizium*, *Penicillium* and *Fusarium*. *Beauveria bassiana* and *Metarhizium anisopliae* have been extensively developed as mycoinsecticides [8]. These species are naturally present in agricultural soils, but the spore numbers in nature are often too low to result in the effective control of pest population outbreaks [9].

Through in-depth studies on the physiology, ecology and molecular biology of EPFs, the effect of applying EPFs to control insects has been significantly improved. Under the premise that pests generally develop resistance, more and more attention has been paid to sustainable development and pollution-free pest management, and researchers

prefer the development and utilization of EPFs [10]. Some fungi have a unique method of infection (they can infect pests through the main body wall), which cannot be replicated by other microbial insecticides. The process of EPFs infecting insects mainly includes host recognition, mechanical destruction, toxin secretion and metabolism interference. The combined effect of various factors leads to the death of the host insects [8]. The host species of EPFs are highly specific, and the host spectrum and virulence of different strains are also quite different. Therefore, the isolation and identification of more strains will help us to enrich the resources of EPF and provide more materials for the development of biological control pesticides using EPF [11].

*Phyllotreta striolata* (Coleoptera: Chrysomelidae) is a prominent pest of *Brassicaceae*, *Solanaceae*, *Cucurbitaceae* and *Leguminosae* vegetables [12–15]. Brassicaceae are important crops in south China [16]. Their management is based on synthetic chemical pesticides, leading to insect resistance [17,18]. Few registered varieties of biopesticides can meet the needs of green prevention and control. EPFs represent the most promising candidates in the integrated pest management (IPM) program approach [19].

Popular EPFs, such as *Beauveria bassiana*, *Metarhizium anisopliae*, *Purpureocillium lilacinum* and *Isaria* (=*Cordyceps*) *javanica*, have been developed as mycopesticides to control agricultural, forest and disease vector pests such as locusts, grubs, aphids, whiteflies, moths, mosquitoes and phytopathogenic nematodes [20,21]. It was found that *B. bassiana* and *M anisopliae* can infect the larvae and adults of *P. striolata* [22,23], but this research is still at the laboratory stage. Because most EPFs are soil-dwelling microbes, investigating soil fungi will be beneficial for exploring new species of EPF resources [24–26].

The Hebei, Henan, Hubei and Hunan provinces have complex and diverse landforms, with a variety of plateaus, mountains, hills, basins and plains, as well as a large latitude span in the Yellow River and Yangtze River basins, which have sufficient water and diverse climate types and are suitable for farming. They are the main agricultural production areas in China and have rich agricultural ecological landscapes. However, the distributions of soil EPFs in these regions are not clear. Therefore, this research aims to investigate the distribution and abundance of EPFs in different soil habits of these Chinese provinces. Moreover, the impacts of human activities and changes in the environment on EPFs are analyzed and discussed. The study of EPFs in the soil of the four areas is beneficial for the exploration of new strains to enrich the diversity of EPFs and for mining highly pathogenic strains.

## 2. Materials and Methods

### 2.1. Soil Sample Collection

The soil samples were collected from different sites (cropland, fallow land, arbor and grassland). The longitude and latitude of each site were recorded by ICEGPS 100C (Shenzhen, China). From each site, approximately 200 g of soil (10~15 cm depth) from three points was collected, mixed and stored in a plastic bag at 4 °C until further use. In total, 226 samples were collected from these sites (Table A1, Appendix A).

### 2.2. Isolation of Fungi from the Soil Samples

The method from our previous work was used to isolate fungal strains from the soil samples [27]. Soil suspensions of 0.02 g/mL were prepared with 0.1% Tween-80 solution; then, 0.1 mL of the suspension was inoculated onto a selective medium (PDA, 0.2 g/L cycloheximide, 0.2 g/L chloramphenicol and 0.013 g/L Bengal red) and cultured at 25 ± 1 °C. When the fungi grew out, a single colony was transferred onto the PDA plate and cultured at 25 ± 1 °C, purified and cultured until a new colony was formed [28].

### 2.3. Identification of Fungal Species and Analysis of Genetic Homology

The identification of fungal isolates was based on the morphological characteristics and similarity of the rDNA-ITS sequences. DNA extraction kits (DP3112, Bio-Teke, Beijing) were used to extract the total DNA from fungal isolates. The primers ITS1 (5′-

TCCGTAGGTGAACCTGCGG-3′) and ITS4 (5′-TCCTCCGCTTATTGATATGC-3′) were used to amplify the ITS region on a T100$^{TM}$ Thermal Cycler (BIO-RAD, Hercules, CA, USA) via a standard PCR cycling protocol (94 °C for 3 min, 94 °C for 30 s, 55 °C for 30 s and 72 °C for 1 min for 33 cycles, then 72 °C for 10 min). The obtained ITS rDNA sequences were submitted to GenBank and compared with similar sequences through the BLAST tool of NCBI. The phylogenetic trees of the fungi were constructed by MEGA X via the statistical method of maximum likelihood, a bootstrap test of 500 replications and the Jukes–Cantor model [29]. The fungal strains are listed in Table 1.

**Table 1.** The information of referred fungal strains.

| Strain/Voucher | GenBank Accession Number | Geographic Origin | Reference |
|---|---|---|---|
| *Acremonium exuviarum* | NR_077167 | Canada | [30] |
| *Acrophialophora nainiana* CBS 417.67 | MK926894 | China | Unpublished |
| *Apiotrichum cacaoliposimilis* ATCC 20505 | NR_154671 | USA | [31] |
| *Arthrographis kalrae* | AB506810 | Japan | [32] |
| *Arthropsis hispanica* CBS 351.92T | HE965759 | Spain | [33] |
| *Aspergillus auricomus* NRRL 391 | NR_135388 | USA | [34] |
| *Aspergillus crustosus* NRRL 4988 | NR_135366 | USA | [34] |
| *Aspergillus fumigatus* ATCC 1022 | NR_121481 | USA | [30] |
| *Aspergillus granulosus* NRRL 1932 | NR_135348 | USA | [30] |
| *Aspergillus niger* ATCC 16888 | NR_111348 | USA | [30] |
| *Aspergillus nomius* NRRL 13137 | NR_121218 | USA | [30] |
| *Aspergillus pseudodeflectus* NRRL 6135 | NR_135372 | USA | [34] |
| *Aspergillus sclerotiorum* NRRL 415 | NR_131294 | USA | [34] |
| *Aspergillus sydowii* CBS 593.65 | NR_131259 | Japan | Unpublished |
| *Aspergillus tanneri* ATCC MYA-4905 | NR_111840 | USA | [30] |
| *Aspergillus terreus* var. *subfloccosus* CBS 117.37 | NR_149331 | Netherlands | [35] |
| *Aspergillus udagawae* CBM FA-0702 | NR_137442 | Japan | Unpublished |
| *Auxarthron alboluteum* UAMH 2846 | NR_111137 | Canada | [30] |
| *Beauveria bassiana* ARSEF 1564 | NR_111594 | USA | [30] |
| *Beauveria bassiana* ARSEF 8187 | HQ444271 | Canada | [30] |
| *Beauveria bassiana* CBS 465.70 | MH859798 | Netherlands | [36] |
| *Beauveria bassiana* CBS 110.25 | MH854802 | Sri Lanka | [36] |
| *Beauveria pseudobassiana* ARSEF 3405 | NR_111598 | USA | [30] |
| *Chloridium aseptatum* MFLU 11-1051 | NR_158365 | China | [37] |
| *Chrysosporium lobatum* CBS 666.78 | NR_111087 | Spain | [30] |
| *Clonostachys grammicospora* CBS 209.93 | NR_137650 | Netherlands | [38] |
| *Clonostachys rosea* f. *catenulata* CBS 154.27 | NR_145021 | Netherlands | [38] |
| *Coniochaeta fasciculata* CBS 205.38 | NR_154770 | Spain | Unpublished |
| *Cordyceps cateniannulata* CBS 152.83 | NR_111169 | Thailand | [30] |
| *Cunninghamella elegans* CBS 160.28 | NR_154747 | China | [39] |
| *Cutaneotrichosporon dermatis* CBS 2043 | NR_130667 | USA | Unpublished |
| *Fusarium falciforme* CBS 475.67 | NR_164424 | Netherlands | [40] |
| *Fusarium keratoplasticum* FRC S-2477 | NR_130690 | USA | [41] |
| *Fusarium solani* CBS 140079 | NR_163531 | Slovenia | [42] |
| *Gongronella butleri* CBS 102.44 | JN206284 | Netherlands | [43] |
| *Gongronella butleri* CBS 157.25 | JN206607 | Netherlands | [43] |
| *Hawksworthiomyces taylorii* CMW 20741 | NR_155176 | South Africa | [44] |
| *Isaria cateniannulata* ARSEF 6242 | GU734760 | Brazil | [45] |
| *Isaria farinosa* ARSEF 4029 | HQ880828 | USA | [46] |
| *Isaria farinosa* CBS 262.58 | AY624179 | Thailand | [47] |
| *Isaria fumosorosea* ARSEF 887 | EU553334 | Brazil | [48] |
| *Isaria fumosorosea* CBS 244.31 | AY624182 | Thailand | [47] |
| *Isaria fumosorosea* CBS 337.52 | EF411219 | Thailand | Unpublished |
| *Isaria javanica* CBS 134.22 | DQ403723 | USA | [49] |

**Table 1.** *Cont.*

| Strain/Voucher | GenBank Accession Number | Geographic Origin | Reference |
|---|---|---|---|
| *Isaria javanica* CHE-CNRCB 303/2 | KM234213 | Mexico | [50] |
| *Lecanicillium coprophilum* CGMCC 3.18986 | NR_163303 | China | [51] |
| *Lecanicillium saksenae* IMI 179841 | NR_111102 | United Kingdom | [30] |
| *Malbranchea aurantiaca* CBS 127.77 | AB040704 | Japan | [52] |
| *Melanoctona tectonae* MFLUCC 12-0389 | NR_154194 | Thailand | Unpublished |
| *Metarhizium anisopliae* CBS 657.67 | MH859066 | Netherlands | [36] |
| *Metarhizium flavoviride* CBS 218.56 | MH857590 | Czech | [36] |
| *Metarhizium marquandii* CBS 282.53 | MH857200 | Netherlands | [36] |
| *Metarhizium marquandii* CBS 182.27 | NR_131994 | Thailand | [47] |
| *Metarhizium carneum* CBS 239.32 | NR_131993 | Thailand | [47] |
| *Metapochonia bulbillosa* 38G272 | EU999952 | USA | [53] |
| *Metapochonia bulbillosa* CBS 145.70 | AJ292397 | UK | [54] |
| *Metapochonia bulbillosa* FKI-4395 | AB709836 | Japan | [55] |
| *Mucor ellipsoideus* ATCC MYA-4767 | NR_111683 | USA | [55] |
| *Nectria mauritiicola* NHRC-FC048 | AJ558115 | Russia | Unpublished |
| *Oidiodendron fuscum* UAMH 8511 | NR_111035 | Canada | [30] |
| *Paecilomyces formosus* CBS 990.73B | NR_149329 | Netherlands | [56] |
| *Paecilomyces variotii* CBS 338.51 | FJ389930 | Netherlands | [56] |
| *Penicillium chrysogenum* CBS 306.48 | NR_077145 | USA | [30] |
| *Penicillium subrubescens* CBS 132785 | NR_111863 | Netherlands | [30] |
| *Penicillium rubens* CBS 319.59 | MH857874 | Netherlands | [30] |
| *Penicillium rubens* CBS 129667 | NR_111815 | Netherlands | [30] |
| *Penicillium guttulosum* NRRL 907 | NR_144820 | USA | [57] |
| *Penicillium citrinum* NRRL 1841 | NR_121224 | USA | [30] |
| *Penicillium mirabile* CBS 624.72 | JN899322 | Netherlands | [58] |
| *Phialophora livistonae* CPC 19433 | NR_111824 | Netherlands | [30] |
| *Purpureocillium lilacinum* CBS 284.36 | NR_111432 | USA | [30] |
| *Purpureocillium lavendulum* FMR 10376 | NR_111433 | Spain | [30] |
| *Simplicillium cylindrosporum* JCM 18169 | NR_111023 | Japan | [30] |
| *Simplicillium minatense* JCM 18176 | NR_111025 | Japan | [30] |
| *Talaromyces pinophilus* CBS 631.66 | NR_111691 | Netherlands | [30] |
| *Talaromyces purpureogenus* CBS 286.36 | NR_121529 | Netherlands | [30] |
| *Talaromyces trachyspermus* CBS 373.48 | NR_147425 | Netherlands | [30] |
| *Talaromyces variabilis* CBS 385.48 | NR_103670 | Netherlands | [30] |
| *Tolypocladium album* CBS 869.73 | NR_155018 | Japan | Unpublished |
| *Trichurus terrophilus* CBS 368.53 | LN850976 | Spain | Unpublished |

### 2.4. Evaluation of the Shannon Evenness Index

The biodiversity of fungi and EPFs in different soils was evaluated using the Shannon Evenness Index (SHEI). The SHEI was calculated via the formula SHEI $= -\sum_i^s (Pi)(\ln Pi)/\ln S$, where s is the total number of species in the sample, *i* is the total number of individuals in one species, *Pi* is the proportion of species in the sample, ln*Pi* is the value of the natural logarithm of *Pi* and *S* is the total number of species.

### 2.5. Bioassay of the Fungal Strains against P. striolata

The isolates of fungal species were subject to a bioassay against *P. striolata* based on the work of [27]. In summary, fungal conidia suspensions of $1.0 \times 10^8$ spores/mL were prepared with 0.02% Tween-80 solution. Spore suspension concentrations of $1.0 \times 10^4$, $1.0 \times 10^5$, $1.0 \times 10^6$, $1.0 \times 10^7$ and $1.0 \times 10^8$ spores/mL were prepared by culturing with a light cycle of 12:12 at 25 °C for 7 days. The population of *P. striolata* was fed with radish lumps, which changed every day. Adults were paralyzed with carbon dioxide and dipped into the conidial suspension for 20 s. The pest populations were surveyed every 24 h after treatment. The 0.02% Tween-80 solution was used as a control group. The experiment was replicated thrice, and 20 adults were used for each treatment.

## 2.6. Scanning Electron Microscopy

The samples were placed in a 2 mL centrifuge tube, fixed with 2.5% glutaraldehyde overnight, washed with physiological saline and dehydrated using a graded series of ethanol; isoamyl acetate was replaced overnight. They were vacuum-dried, fixed onto the platform and then coated with platinum with an ion coater before being observed using a scanning electron microscope.

## 2.7. Statistical Analysis

Analyses of the bioassay data were carried out using IBM SPSS Statistics version 20.0 (IBM Corp., Armonk, NY, USA). The data were expressed as mean $\pm$ SD and were subjected to one-way ANOVA, followed by Duncan's multiple range test (DMRT). Significant differences were accepted at $p < 0.05$.

## 3. Results

### 3.1. EPF Species Diversity in the Soils of China

In total, 302 fungal isolates were purified. Among these, 188 EPF isolates were identified as belonging to 11 genera according to the morphological and molecular analyses. *Purpureocillium lavendulum*, with 69 isolates, was the dominant species, and the congeneric species *Purpureocillium lilacinum* had only 13 isolates (Figure 1, Table A1). The genus *Metarhizium* had three species—*M. anisopliae*, *M. marquandii* and *M.* sp.—for which 49, 33 and 17 isolates, respectively, were obtained (Figure 1, Table A1). *Penicillium* had six species—*Penicillium subrubescens*, *Penicillium guttulosum*, *Penicillium rubens*, *Penicillium chrysogenum*, *Penicillium citrinum* and *Penicillium mirabile*—with 12, 2, 3, 1, 11 and 6 isolates found, respectively (Figure 2, Table A1). *Aspergillus* had 12 species (Figure 2, Table A1). *Talaromyces* had four species (Figure 3, Table A1), and both *Beauveria* and *Isaria* had three species each (Figure 3, Table A1). Both *Lecanicillium* and *Simplicillium* had four species each (Figure 4, Table A1). *Fusarium*, *Coniochaeta* and *Clonostachys* each had six species (Figure 4, Table A1). Other species with one to four isolates were identified as *Tolypocladium album*, *Acremonium exuviarum*, *Acrophialophora nainiana*, *Nectria mauritiicola*, *Hawksworthiomyces taylorii*, *Chloridium aseptatum*, *Trichurus terrophilus*, *Chrysosporium lobatum*, *Arthropsis hispanica*, *Malbranchea aurantiaca*, *Auxarthron alboluteum*, *Arthrographis kalrae*, *Melanoctona tectonae*, *Phialophora livistonae*, *Xenopolyscytalum pinea*, *Oidiodendron fuscum*, *Cutaneotrichosporon dermatis*, *Apiotrichum cacaoliposimilis*, *Mucor ellipsoideus*, *Gongronella butleri* and *Cunninghamella elegans*. (Figure 5, Table A1). The other 73 isolates have not been classified yet. Obviously, *Purpureocillium lavendulum*, *M. anisopliae*, *M. marquandii*, *Purpureocillium lilacinum* and *B. bassiana* were the most abundant EPF species.

### 3.2. Distribution of Soil EPF in Different Regions

There were different numbers and isolating rates of EPFs in different regions. Compared with the average fungal isolating rates of 83.70% and 61.92% in all fungi and EPFs, Henan had the highest rate of >90% (Table 2). However, the Shannon Evenness Index indicated that Hubei and Hunan were districts with the highest EPF biodiversity, while Hunan and Hebei had the EPF biodiversity with SHEI values of 0.87 and 0.88, respectively (Table 2).

### 3.3. The Biodiversity of Soil EPF in Different Environments

There were different numbers and isolating rates of EPF in Central China. Compared with the average fungal isolating rates of 87.42% and 61.16% for all fungi and EPFs, cropland samples had higher rates of >69% (Table 3). However, the SHEI indicated that cropland had the lowest EPF biodiversity, while fallow land samples had the most abundant EPF biodiversity (Table 3).

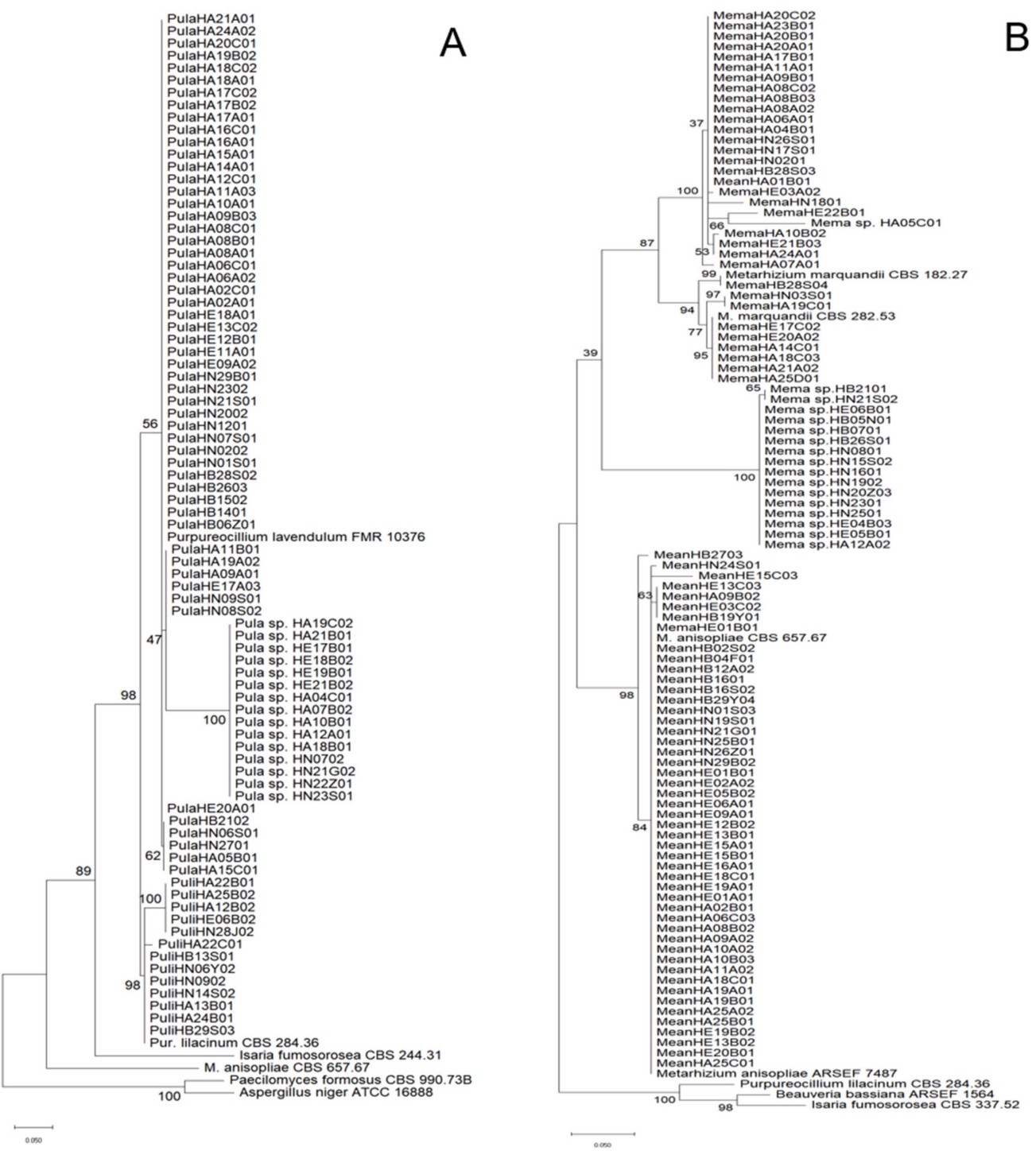

**Figure 1.** Phylogenetic tree of *Purpureocillium* spp. (**A**) and *Metarhizium* spp. (**B**) isolates.

*3.4. The Pathogenicity of Fungal Isolates against P. striolata*

Forty-seven isolates were subjected to a bioassay against *P. striolata*. The results indicate that *I. javanica* (IsjaHN3002) had the highest mortality, and *Aspergillus* spp., *Fusarium falciforme*, *Lecanicillium* spp., *Metarhizium* spp. and *Talaromyces* spp. all had obvious pathogenicity against *P. striolata* (Table 4).

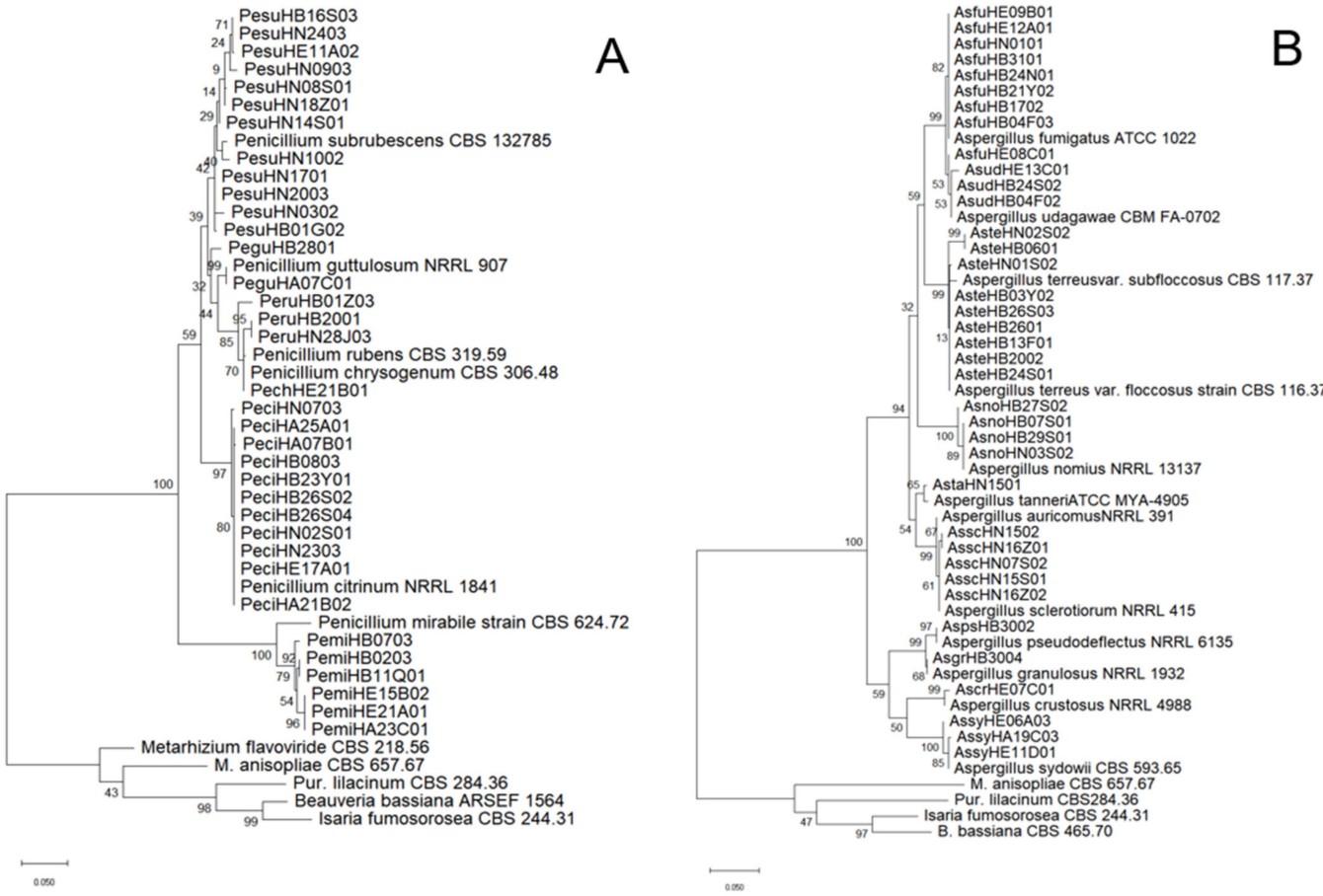

**Figure 2.** Phylogenetic tree of *Penicillium* spp. (**A**) and *Aspergillus* spp. (**B**) isolates.

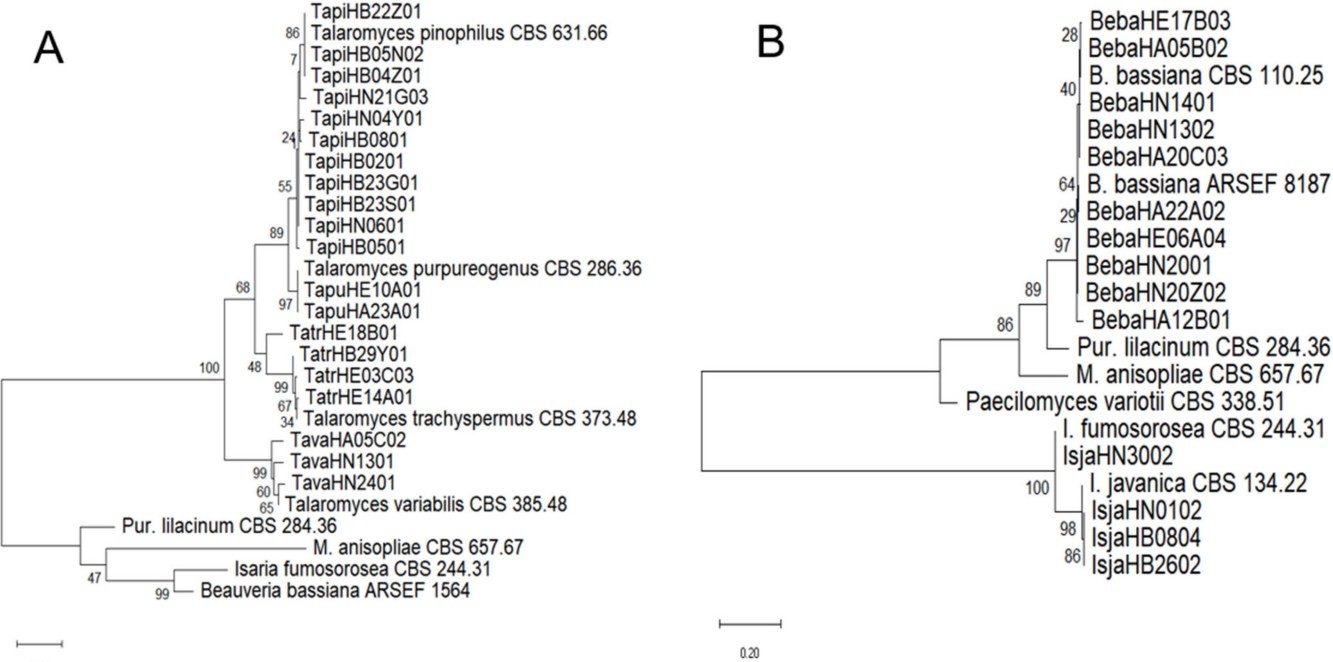

**Figure 3.** Phylogenetic tree of *Talaromyces* spp. (**A**) and *Beauveria/Isaria* (**B**) isolates.

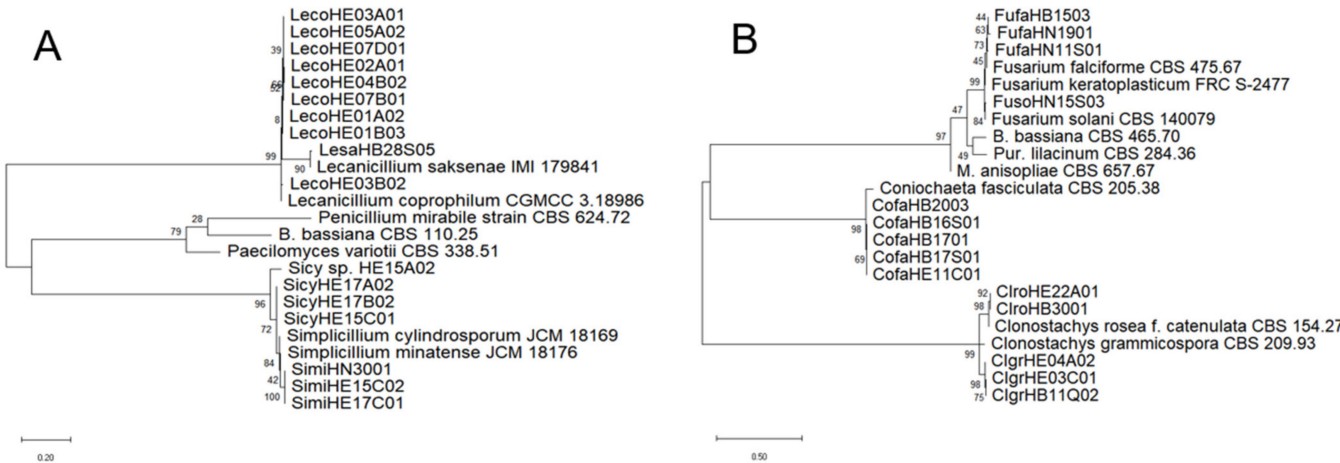

**Figure 4.** Phylogenetic tree of *Lecanicillium*/*Simplicillium* spp. (**A**) and *Fusarium*/*Coniochaeta*/*Clonostachys* (**B**) isolates.

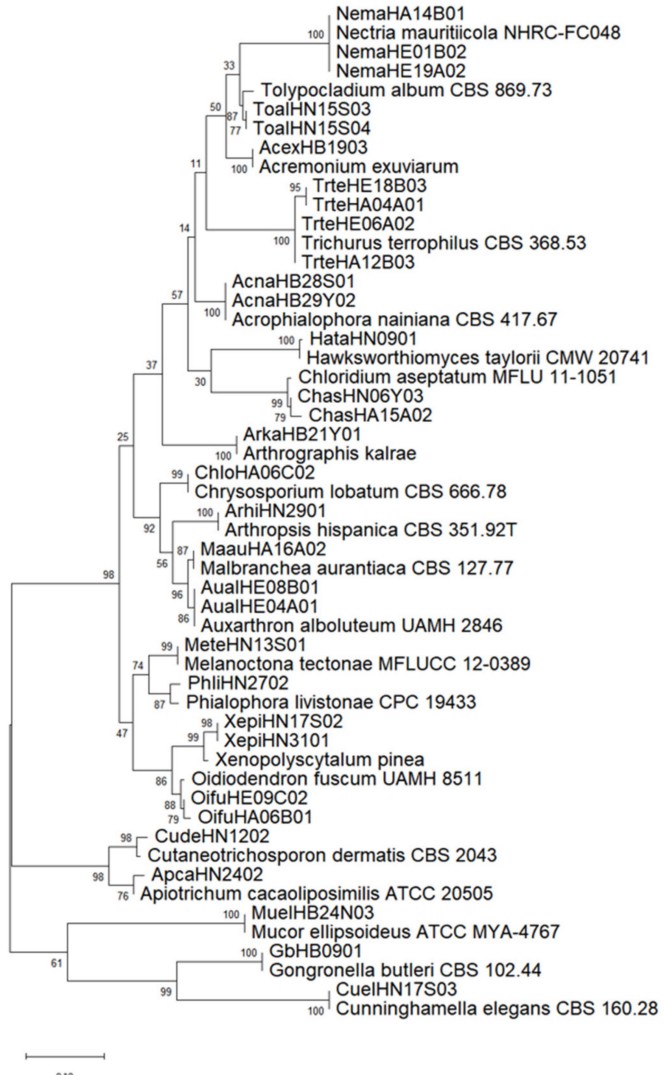

**Figure 5.** Phylogenetic tree of other isolates.

**Table 2.** The fungi isolation and biodiversity of different regions.

| Region | Soil Sample Numbers | Isolate Number | | Isolation Rate (%) | | EPF Species | SHEI |
|---|---|---|---|---|---|---|---|
| | | Fungi | EPF | Fungi | EPF | | |
| Hunan | 54 | 97 | 39 | 85.19 | 62.96 | 9 | 0.87 |
| Hubei | 50 | 58 | 24 | 80.00 | 40.00 | 8 | 0.88 |
| Henan | 63 | 73 | 83 | 98.41 | 90.48 | 7 | 0.84 |
| Hebei | 59 | 74 | 42 | 71.19 | 54.24 | 8 | 0.78 |
| Total | 226 | 302 | 188 | 83.70 * | 61.92 * | 11 | – |

* The mean isolation rate (%) of all regions. EPF: Entomopathogenic fungi; SHEI: Shannon evenness index.

**Table 3.** The fungi isolation and biodiversity of different samples.

| Region | Soil Sample Numbers | Isolate Number | | Isolation Rate (%) | | EPF Species | SHEI |
|---|---|---|---|---|---|---|---|
| | | Fungi | EPF | Fungi | EPF | | |
| Arbor | 64 | 79 | 45 | 79.69 | 59.38 | 9 | 0.86 |
| Crop | 85 | 114 | 83 | 84.71 | 69.41 | 9 | 0.76 |
| Fallow land | 9 | 12 | 6 | 100.00 | 55.56 | 6 | 1.00 |
| Grass | 68 | 97 | 54 | 85.29 | 60.29 | 9 | 0.81 |
| Total | 226 | 302 | 188 | 87.42 * | 61.16 * | 11 | – |

* The mean isolation rate (%) of all vegetation types sampled. EPF: Entomopathogenic fungi, SHEI: Shannon evenness index. Arbor: lands covered with arbor forests; cropland: farming lands planted with crops; fallow land: farming lands with no crops; grass: lands covered with grass.

**Table 4.** The pathogenicity of fungal isolates against adults of *P. striolata*.

| Isolated Strain | Species | Adjusted Mortality (%) |
|---|---|---|
| ApcaHN2402 | *Apiotrichum cacaoliposimilis* | 12.96 ± 0.56 |
| AsgrHB3004 | *Aspergillus granulosus* | 26.19 ± 0.24 |
| AsnoHB27S02 | *Aspergillus nomius* | 21.05 ± 0.72 |
| AsnoHN03S02 | *Aspergillus nomius* | 12.96 ± 0.57 |
| AspsHB3002 | *Aspergillus pseudodeflectus* | 21.88 ± 0.85 |
| AsscHN1502 | *Aspergillus sclerotiorum* | 7.55 ± 0.35 |
| AssyHE06A03 | *Aspergillus sydowii* | 19.30 ± 0.38 |
| AstaHN1501 | *Aspergillus tanneri* | 45.24 ± 0.39 |
| AsteHN01S02 | *Aspergillus terreus* | 15.71 ± 0.73 |
| AsudHE13C01 | *Aspergillus udagawae* | 16.00 ± 0.44 |
| BebaHA22A02 | *Beauveria bassiana* | 10.70 ± 0.19 |
| ChasHA15A02 | *Chloridium aseptatum* | 1.28 ± 0.51 |
| CudeHN1202 | *Cutaneotrichosporon dermatis* | 2.83 ± 0.10 |
| FufaHN1901 | *Fusarium falciforme* | 28.07 ± 0.68 |
| IsjaHB2602 | *Isaria javanica* | 9.52 ± 0.30 |
| IsjaHN3002 | *Isaria javanica* | 67.86 ± 0.61 |
| LecoHE07B01 | *Lecanicillium coprophilum* | 18.18 ± 0.33 |
| LesaHB28S05 | *Lecanicillium saksenae* | 26.32 ± 0.45 |
| MeanHE15B01 | *Metarhizium anisopliae* | 23.56 ± 0.37 |
| MeanHE20B01 | *Metarhizium anisopliae* | 19.62 ± 0.45 |
| MemaHA24A01 | *Metarhizium marquandii* | 4.55 ± 0.42 |
| MemaHN26S01 | *Metarhizium marquandii* | 22.81 ± 0.91 |
| Mema sp. HN2501 | *Metarhizium marquandii* | 5.63 ± 0.41 |
| MeteHN13S01 | *Melanoctona tectonae* | 15.00 ± 1.12 |
| MuelHB24N03 | *Mucor ellipsoideus* | 15.00 ± 0.60 |
| NemaHA14B01 | *Nectria mauritiicola* | 16.33 ± 0.30 |
| OifuHA06B01 | *Oidiodendron fuscum* | 1.85 ± 0.19 |
| PeciHA25A01 | *Penicillium citrinum* | 9.74 ± 0.29 |
| PesuHN1002 | *Penicillium subrubescens* | 6.67 ± 0.27 |
| PhliHN2702 | *Phialophora livistonae* | 16.33 ± 0.57 |

**Table 4.** *Cont.*

| Isolated Strain | Species | Adjusted Mortality (%) |
|---|---|---|
| PulaHA08C01 | *Purpureocillium lavendulum* | 3.92 ± 0.30 |
| SicyHE17A02 | *Simplicillium cylindrosporum* | 15.00 ± 0.27 |
| SimiHE17C01 | *Simplicillium minatense* | 8.16 ± 0.23 |
| TapiHB23G01 | *Talaromyces pinophilus* | 7.02 ± 0.16 |
| TapiHB23S01 | *Talaromyces pinophilus* | 31.58 ± 0.32 |
| TatrHE03C03 | *Talaromyces trachyspermus* | 11.11 ± 0.20 |
| TatrHE14A01 | *Talaromyces trachyspermus* | 6.56 ± 0.44 |
| TatrHE18B01 | *Talaromyces trachyspermus* | 8.33 ± 0.21 |
| ToalHN15S03 | *Tolypocladium album* | 11.67 ± 0.65 |
| HA13B02 | – | 2.27 ± 0.31 |
| HA17C01 | – | 11.43 ± 0.18 |
| HB3003 | – | 23.81 ± 0.34 |
| HB3102 | – | 8.87 ± 0.69 |
| HE07A01 | – | 14.81 ± 0.32 |
| HN06Y05 | – | 14.81 ± 0.41 |
| HN20Z01 | – | 7.41 ± 0.23 |
| HN28J01 | – | 1.96 ± 0.22 |
| Control | – | 3.33 ± 0.45 |

Data in the table are mean values ± standard error. For each test, 20 *P. striolata* samples were used in each treatment, and the concentration of the fungal spore suspension was $1.0 \times 10^8$ spores/mL. The experiment was repeated three times.

### 3.5. The Pathogenicity of I. javanica against P. striolata

According to the results shown in Table 5, the number of muscardine cadavers increased with the spore concentration. The lethal rate of $1.0 \times 10^8$ spores/mL spore suspension treatment group was as high as 80%. When the spore concentration was lower than $1.0 \times 10^6$ spores/mL, no hyphae were observed on the body wall of *P. striolata* in the first 3 days. There was no significant difference in the rate of zombies in the groups treated with spore suspensions at concentrations of $1.0 \times 10^4$ and $1.0 \times 10^5$, $1.0 \times 10^6$ spores/mL in the first 3 days, but there was a significant difference in the rate of zombies in the group treated with spore suspensions with concentrations of $1.0 \times 10^7$ and $1.0 \times 10^8$ spores/mL in the first 3 days. After the seventh day, the differences among the treatment groups were revealed. Compared with other treatment groups, there was a significant difference in the lethal rate of the spore suspension with a concentration of $1.0 \times 10^8$ spores/mL.

**Table 5.** Pathogenicity of *I. javanica* in different concentrations against *P. striolata*.

| Concentration (Spores/mL) | Accumulated Mortality (%) | | | Muscardine Cadaver Rate (%) | | |
|---|---|---|---|---|---|---|
| | 1 d | 3 d | 7 d | 1 d | 3 d | 7 d |
| CK | 3.33 ± 2.36 c | 6.67 ± 2.36 c | 10.00 ± 4.08 d | 0 | 0 | 0 |
| $1.0 \times 10^4$ | 10.00 ± 4.08 bc | 18.33 ± 2.36 b | 40.00 ± 4.08 c | 0 | 0 | 6.67 ± 4.71 c |
| $1.0 \times 10^5$ | 15.00 ± 0 ab | 23.33 ± 2.36 b | 41.67 ± 2.36 c | 0 | 0 | 11.67 ± 4.71 c |
| $1.0 \times 10^6$ | 13.33 ± 6.23 abc | 20.00 ± 4.08 b | 53.33 ± 4.71 b | 0 | 1.67 ± 2.36 b | 21.67 ± 6.23 b |
| $1.0 \times 10^7$ | 21.67 ± 8.5 a | 33.33 ± 6.23 a | 61.67 ± 4.71 b | 0 | 3.33 ± 2.36 b | 26.67 ± 2.36 b |
| $1.0 \times 10^8$ | 23.33 ± 2.36 a | 36.67 ± 4.71 a | 80.00 ± 4.08 a | 0 | 8.33 ± 2.36 a | 41.67 ± 2.36 a |

Muscardine cadavers were dead insects that grew mycelium. Accumulated mortality includes all dead insects. The data are the mean ± SE on different days after treatment. Different letters indicate a significant difference ($p < 0.05$) determined by DMRT.

### 3.6. Scanning Electron Microscopy Observations of Infection Process of I. javanica

The results showed that the attachment of conidia of *I. javanica* to different parts of the body surface was very different. After 2 h, the attachment of conidia was observed. No attachment of conidia was found on the head, abdomen, shard or other smooth surfaces. The conidia were mainly attached to the bristly areas and internodes such as the antennae,

foot joints, chest and chest feet. The most densely attached site was the intersegmental membrane of the chest feet, followed by the foot joints (Figure 6).

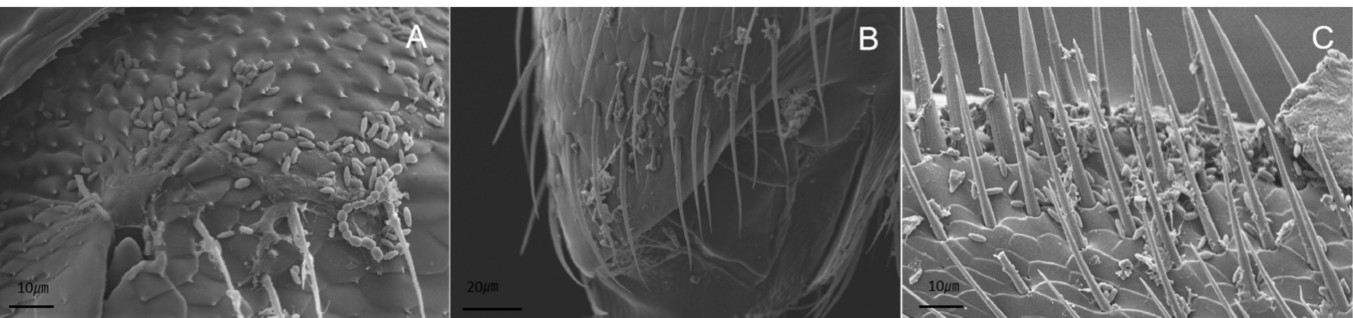

**Figure 6.** The attachment of conidia of *I. javanica* on the body surface after inoculation for 2 h. (**A**) Chest internode; (**B**) hind foot internode; (**C**) foot end.

After 12 h of inoculation, some conidia began to germinate, forming short germ tubes at the top. Twenty-four hours after infection, the top of the germ tube expanded to form an appressorium and continued in the direction of the intersegmental membrane, forming tendrils (Figure 7A–C) and looking for a suitable invasion site. The germ tube could also directly invade the body wall (Figure 7D). At 48 h, hyphae began to grow between the foot internode, and new conidiophores and conidia sprouted (Figure 8A). Next, 48–72 h after inoculation, the surface of the insect body was gradually covered by mycelia until it was completely covered (Figure 8B–D). Through stereoscopic observation, the mycelia were observed to grow from the body surface on the third day, and then the mycelium coverage increased day by day (Figure 9), while the control group never experienced mycelial growth.

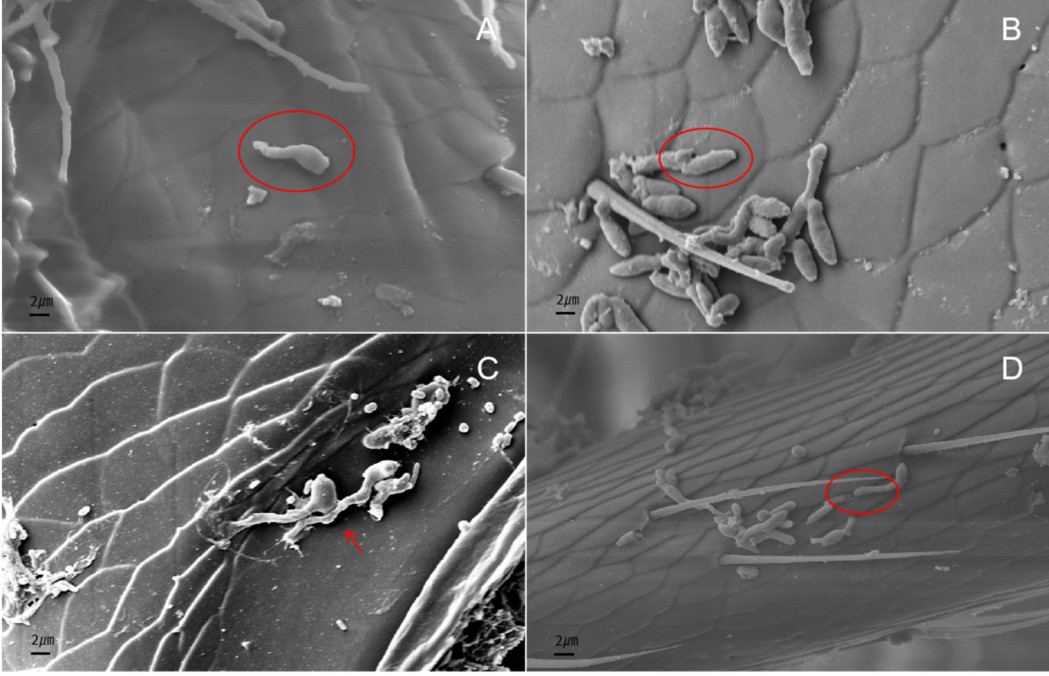

**Figure 7.** SEM observations of the inoculation of *I. javanica* for 12–24 h. (**A**) Spores germinate to form a short dental canal; (**B**) apical expansion to form an adherent cell; (**C**) adnexal extension; (**D**) bud tube invades the body wall.

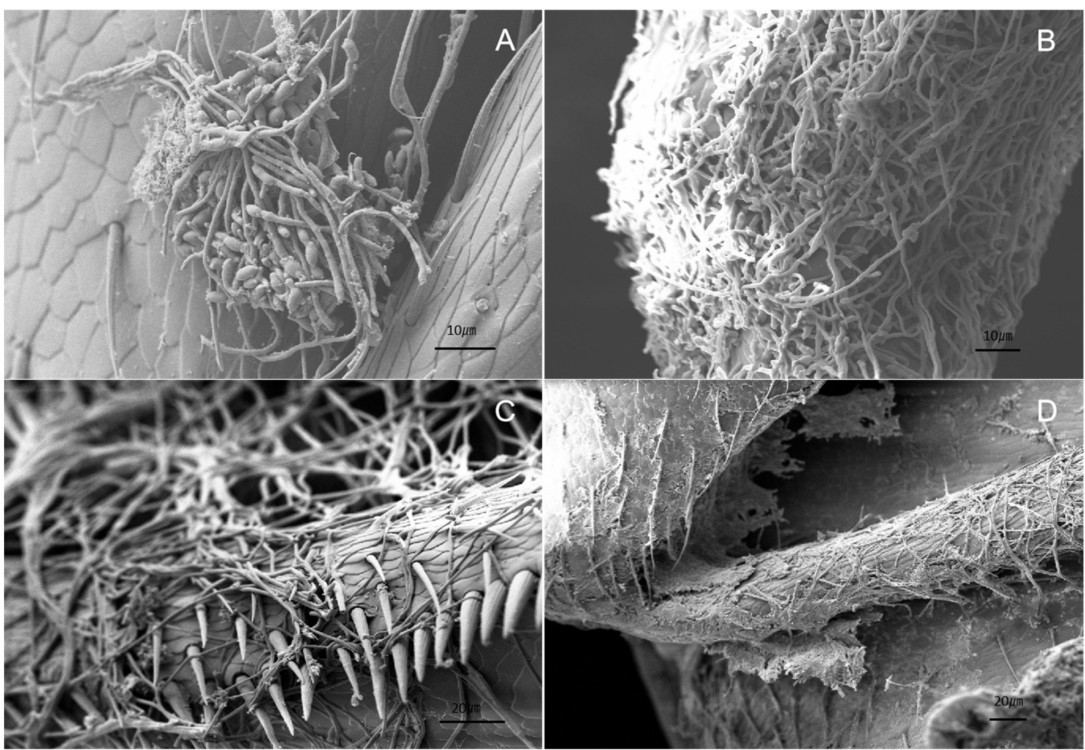

**Figure 8.** SEM observations of the inoculation of *I. javanica* for 48–72 h. (**A**) At 48 h, hyphae grew between the foot nodes, and new conidiophores and new conidia were formed. (**B**) At 60 h, conidia germination in vitro produced new hyphae covering the hindfoot. (**C**) At 72 h, the end of the foot was covered with mycelia. (**D**) At 72 h, the hind foot was covered with mycelia.

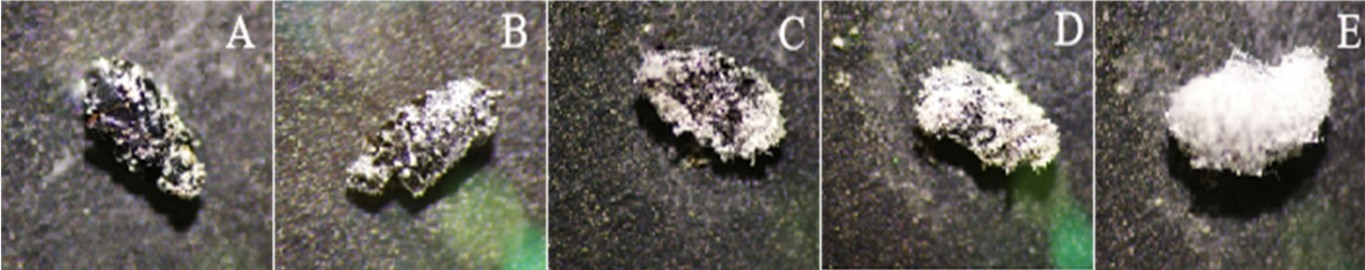

**Figure 9.** Mycelial growth of *Phyllotreta striolata* infected by *I. javanica*. (**A**) 3 d; (**B**) 4 d; (**C**) 5 d; (**D**) 6 d; (**E**) 7 d.

## 4. Discussion

This study surveyed the EPF distribution at a broad scale in China. ITS sequences are small and easy to analyze and have been widely used in the phylogenetic analysis of different fungal species, but their accuracy is controversial. Therefore, the identification of the fungal species in this study has some defects. Undoubtedly, our results initially provide a large amount of information about the soil fungi in these areas. Moreover, the results indicate that the soil environment strongly impacts the distribution of EPFs. Compared to arbor and non-cultivated land, the cropland samples had fewer EPFs. The isolation rate of EPFs was not high, which showed that soil fungi were not abundant in these areas and that the sampling and isolation methods also affected the isolation of fungi. The EPF diversity may be affected by the use of fungicides in croplands. China is a heavy consumer of pesticides, and a large number of broad-spectrum fungicides such as carbendazim, chlorothalonil and azoxystrobin, etc., are sprayed on croplands and probably inhibit fungi [59,60].

EPFs can parasitize insects and cause insect diseases, including some obligate parasitism that may not cause insect death but that can reduce the vitality of the host insects and weaken them [61] or affect insect spawning [62]; as such, when using EPFs, we can observe changes in the behavior of host insects [63,64]. Some studies have suggested that insects can actively identify fungi, with the target location being the cell wall of the fungi, while the fungi will take a series of measures to evade the host's defenses in the face of insect recognition [65]. Therefore, the invasion of host insects by EPF is a process of mutual influence and interaction [66]. As a result, the body surface of *P. striolata* may be able to recognize *I. javanica*, and the resistance and defense of *I. javanica* may also take measures to promote the germination of conidia in advance. In view of this fact, we can further explore what receptor binds the cell wall of conidia of *I. javanica* to produce signal molecules and promote spore germination, determining the factors promoting spore germination and improving pathogenicity.

Through scanning electron microscope observation, 12 h after infection with *I. javanica*, some conidia began to germinate, as shown in Figure 7. After 24 h of infection, only some scattered spores germinated. Because of the hard shell and dense structure on the body surface, the structure of the body wall varies greatly in different parts, and the outer skin has hydrophobic components. However, in tests of the bioactivity of different concentrations of spore suspensions against *P. striolata*, it was found that the spore suspension concentration of *I. javanica* had a stimulative effect on the production of zombies. This may be the QS phenomenon observed in *I. javanica*, which refers to a change in the physiological and biochemical characteristics of the microbial population in the process of its growth due to an increase in the population density, showing the characteristics of a small number of bacteria or a single bacterium. Cells use the QS mechanism to carry out cell-to-cell communication so that they can coordinate in a complex environment, and their "team combat ability" better ensures that the whole population survives. At present, the study of QSM is mainly focused on bacteria, and QSM has also been reported in related fungi [67]. In recent years, more reports have confirmed that fungi have QSM [68,69] and have QSM pheromones that are similar to the bacterial regulation of the physiological behavior of fungi [70–72]. However, in-depth studies of fungal QSM have not been carried out. Therefore, in the production of fungicidal insecticides using *I. javanica*, we can choose the appropriate formulation or use new production technology to help *I. javanica* survive in the form of sporangia, and it can also attach to the body surface after application to invade the body faster and improve its pathogenicity.

Several species have not been reported as EPF, namely *Aspergillus*, *Lecanicillium*, *Monascus*, *Talaromyces* and *Fusarium*. Their pathogenicity against *P. striolata* was discovered, and their potential for pest control deserves further research. Our experiment will provide new insight into the distribution characteristics of EPF and the conservation of their biodiversity.

## 5. Conclusions

In conclusion, 188 EPF isolates were identified from 226 soil samples, and the amount and types of fungi in the soil varied by region and vegetation type. *Metarhizium*, with 89 isolates, was recognized as the dominant EPF species, whereas *Purpureocillium* and *Beauveria* (respectively with 81 and 11 isolates) were the richer genera. Finally, it was first reported that *I. javanica* had pathogenicity against *P. striolata*, and we described its infection process.

**Author Contributions:** K.Z. and X.Z. completed most of the experiments, including the collection of the soil samples, the isolation and identification of the fungi strains and the bioassay and data analysis. Q.H. designed the experiments and collected partial soil samples. K.Z. and Q.W. wrote the article. All authors have read and agreed to the published version of the manuscript.

**Funding:** This project was supported by Guangdong Province Science and Technology Project (2016B020234005) and the National Natural Science Foundation of China (31572053).

**Institutional Review Board Statement:** Not applicable.

**Informed Consent Statement:** Not applicable.

**Data Availability Statement:** Publicly available datasets were analyzed in this study. These data can be found here: https://www.ncbi.nlm.nih.gov/nuccore/?term=OM372687:OM373035[accn], submission ID SUB9030162; accessed on 25 January 2022.

**Conflicts of Interest:** The authors declare no conflict of interest.

## Appendix A

**Table A1.** The information of the soil samples collected and fungal isolates.

| NO. | Address | Site — Latitude and Longitude | Sample Environment | Isolate | GenBank Access No. | Species |
|-----|---------|-------------------------------|--------------------|---------|--------------------|---------|
| HB01 | Xianning, Hubei | 29.267 N, 113.746 E | Fallow land | HB01Z01 | – | |
| | | | | HB01Z02 | – | |
| | | | | PeruHB01Z03 | OM372687 | *Penicillium rubens* |
| | | | | HB01Z04 | – | |
| | | | Crop | HB01G01 | – | |
| | | | | PesuHB01G02 | OM372688 | *Penicillium subrubescens* |
| HB02 | Xianning, Hubei | 29.568 N, 114.193 E | Crop | – | – | |
| | | | Arbor | HB02S01 | – | |
| | | | | MeanHB02S02 | OM372689 | *Metarhizium anisopliae* |
| | | | Grass | TapiHB0201 | OM372690 | *Talaromyces pinophilus* |
| | | | | HB0202 | – | |
| | | | | PemiHB0203 | OM372691 | *Penicillium mirabile* |
| HB03 | Daye, Hubei | 29.973 N, 114.667 E | Crop | HB03Y01 | – | |
| | | | | AsteHB03Y02 | OM372692 | *Aspergillus terreus* |
| | | | | HB03Y03 | – | |
| | | | Grass | HB0301 | – | |
| HB04 | Huanggang, Hubei | 30.372 N, 115.161 E | Fallow land | TapiHB04Z01 | OM372693 | *Talaromyces pinophilus* |
| | | | Crop | MeanHB04F01 | OM372694 | *Metarhizium anisopliae* |
| | | | | AsudHB04F02 | OM372695 | *Aspergillus udagawae* |
| | | | | AsfuHB04F03 | OM372696 | *Aspergillus fumigatus* |
| HB05 | Xinzhou, Hubei | 30.863 N, 114.881 E | Crop | Mema sp. HB05N01 | OM372697 | *Metarhizium marquandii* |
| | | | | TapiHB05N02 | OM372698 | *Talaromyces pinophilus* |
| | | | Grass | TapiHB0501 | OM372699 | *Talaromyces pinophilus* |
| HB06 | Huanggang, Hubei | 31.257 N, 115.056 E | Arbor | HB06S01 | – | |
| | | | Fallow land | PulaHB06Z01 | OM372700 | *Purpureocillium lavendulum* |
| | | | Grass | AsteHB0601 | OM372701 | *Aspergillus terreus* |
| HB07 | Wuhan, Hubei | 30.887 N, 114.462 E | Grass | Mema sp. HB0701 | OM372702 | *Metarhizium marquandii* |
| | | | | HB0702 | – | |
| | | | | PemiHB0703 | OM372703 | *Penicillium mirabile* |
| | | | Arbor | AsnoHB07S01 | OM372704 | *Aspergillus nomius* |
| HB08 | Xiaogan, Hubei | 31.030 N, 113.938 E | Crop | – | – | |
| | | | Grass | TapiHB0801 | OM372705 | *Talaromyces pinophilus* |
| | | | | HB0802 | – | |
| | | | | PeciHB803 | OM372706 | *Penicillium citrinum* |
| | | | | IsjaHB0804 | OM372707 | *Isaria javanica* |
| HB09 | Xiaogan, Hubei | 31.325 N, 113.580 E | Fallow land | GobuHB0901 | OM372708 | *Gongronella butleri* |
| | | | | HB0902 | – | |
| | Suizhou, Hubei | 31.665 N, 113.269 E | Arbor | – | – | |
| | | | Grass | – | – | |
| HB11 | Xiangyang, Hubei | 31.948 N, 112.929 E | Crop | HB11Y01 | – | |
| | | | | PemiHB11Q01 | OM372709 | *Penicillium mirabile* |
| | | | | ClgrHB11Q02 | OM372710 | *Clonostachys grammicospora* |
| HB12 | Xiangyang, Hubei | 32.178 N, 112.211 E | Grass | HB12A01 | – | |
| | | | | MeanHB12A02 | OM372711 | *Metarhizium anisopliae* |
| | | | Arbor | – | | |
| HB13 | Xiangyang, Hubei | 32.307 N, 111.614 E | Crop | AsteHB13F01 | OM372712 | *Aspergillus terreus* |
| | | | | HB13F02 | – | |
| | | | Arbor | PuliHB13S01 | OM372713 | *Purpureocillium lilacinum* |

**Table A1.** *Cont.*

| NO. | Address | Latitude and Longitude | Sample Environment | Isolate | GenBank Access No. | Species |
|---|---|---|---|---|---|---|
| | | **Site** | | | | |
| HB14 | Shiyan, Hubei | 32.502 N, E111.100 E | Grass | PulaHB1401 | OM372714 | *Purpureocillium lavendulum* |
| | | | Arbor | HB14S01 | – | |
| HB15 | Shiyan, Hubei | 32.020 N, 110.679 E | Fallow land | – | | |
| | | | Grass | HB1501 | – | |
| | | | | PulaHB1502 | OM372715 | *Purpureocillium lavendulum* |
| | | | | FufaHB1503 | OM372716 | *Fusarium falciforme* |
| HB16 | Shennongjia, Hubei | 31.823 N, 110.508 E | Grass | MeanHB1601 | OM372717 | *Metarhizium anisopliae* |
| | | | Arbor | CofaHB16S01 | OM372718 | *Coniochaeta fasciculata* |
| | | | | MeanHB16S02 | OM372719 | *Metarhizium anisopliae* |
| | | | | PesuHB16S03 | OM372720 | *Penicillium subrubescens* |
| HB17 | Shennongjia, Hubei | 31.514 N, 110.338 E | Grass | CofaHB1701 | OM372721 | *Coniochaeta fasciculata* |
| | | | | AsfuHB1702 | OM372722 | *Aspergillus fumigatus* |
| | | | Arbor | CofaHB17S01 | OM372723 | *Coniochaeta fasciculata* |
| HB18 | Yichang, Hubei | 31.266 N, 110.686 E | Grass | HB1801 | – | |
| HB19 | Enshi, Hubei | 30.007 N, 110.377 E | Grass | HB1901 | – | |
| | | | | HB1902 | – | |
| | | | | AcexAB1903 | OM372724 | *Acremonium exuviarum* |
| | | | Crop | MeanHB20Y01 | OM372725 | *Metarhizium anisopliae* |
| HB20 | Enshi, Hubei | 30.556 N, 109.889 E | Grass | PeruHB2001 | OM372726 | *Penicillium rubens* |
| | | | | AsteHB2002 | OM372727 | *Aspergillus terreus* |
| | | | | CofaHB2003 | OM372728 | *Coniochaeta fasciculata* |
| | | | Arbor | HB20S01 | – | |
| | | | | HB20S02 | – | |
| HB21 | Yichang, Hubei | 30.615 N, 110.513 E | Grass | Mema sp. HB2101 | OM372729 | *Metarhizium marquandii* |
| | | | | PulaHB2102 | OM372730 | *Purpureocillium lavendulum* |
| | | | Crop | ArkaHB21Y01 | OM372731 | *Arthrographis kalrae* |
| | | | | AsfuHB21Y02 | OM372732 | *Aspergillus fumigatus* |
| HB22 | Yichang, Hubei | 30.582 N, 111.028 E | Fallow land | TapiHB22Z01 | OM372733 | *Talaromyces pinophilus* |
| | | | Crop | HB22Y01 | – | |
| | | | | HB22Y02 | – | |
| HB23 | Yichang, Hubei | 30.688 N, 111.517 E | Crop | PeciHB23Y01 | OM372734 | *Penicillium citrinum* |
| | | | Orchard | TapiHB23G01 | OM372735 | *Talaromyces pinophilus* |
| | | | Arbor | TapiHB23S01 | OM372736 | *Talaromyces pinophilus* |
| HB24 | Jingmen, Hubei | 30.904 N, 112.185 E | Arbor | AsteHB24S01 | OM372737 | *Aspergillus terreus* |
| | | | | AsudHB24S02 | OM372738 | *Aspergillus udagawae* |
| | | | Crop | AsfuHB24N01 | OM372739 | *Aspergillus fumigatus* |
| | | | | HB24N02 | – | |
| | | | | MuelHB24N03 | OM372740 | *Mucor ellipsoideus* |
| | | | | HB24N04 | – | |
| HB25 | Jingmen, Hubei | 30.991 N, 112.854 E | Arbor | HB25S01 | – | |
| | | | Grass | – | | |
| HB26 | Xiaogan, Hubei | 30.868 N, 113.576 E | Arbor | Mema sp.HB26S01 | OM372741 | *Metarhizium marquandii* |
| | | | | PeciHB26S02 | OM372742 | *Penicillium citrinum* |
| | | | | AsteHB26S03 | OM372743 | *Aspergillus terreus* |
| | | | | PeciHB26S04 | OM372744 | *Penicillium citrinum* |
| | | | Grass | AsteHB2601 | OM372745 | *Aspergillus terreus* |
| | | | | IsjaHB2602 | OM372746 | *Isaria javanica* |
| | | | | PulaHB2603 | OM372747 | *Purpureocillium lavendulum* |
| HB27 | Wuhan, Hubei | 30.478 N, 113.874 E | Arbor | HB27S01 | – | |
| | | | | AsnoHB27S02 | OM372748 | *Aspergillus nomius* |
| | | | Grass | HB2701 | – | |
| | | | | HB2702 | – | |
| | | | | MeanHB2703 | OM372749 | *Metarhizium anisopliae* |
| HB28 | Xiantao, Hubei | 30.350 N, 113.424 E | Grass | PeguHB2801 | OM372750 | *Penicillium guttulosum* |
| | | | Arbor | AcnaHB28S01 | OM372751 | *Acrophialophora nainiana* |
| | | | | PulaHB28S02 | OM372752 | *Purpureocillium lavendulum* |
| | | | | MemaHB28S03 | OM372753 | *Metarhizium marquandii* |
| | | | | MemaHB28S04 | OM372754 | *Metarhizium marquandii* |
| | | | | LesaHB28S05 | OM372755 | *Lecanicillium saksenae* |

**Table A1.** *Cont.*

| | Site | | | Isolate | GenBank Access No. | Species |
|---|---|---|---|---|---|---|
| NO. | Address | Latitude and Longitude | Sample Environment | | | |
| HB29 | Qianjiang, Hubei | 30.373 N, 112.889 E | Crop | TatrHB29Y01 | OM372756 | *Talaromyces trachyspermus* |
| | | | | AcnaHB29Y02 | OM372757 | *Acrophialophora nainiana* |
| | | | | HB29Y03 | – | |
| | | | Arbor | MeanHB29Y04 | OM372758 | *Metarhizium anisopliae* |
| | | | | AsnoHB29S01 | OM372759 | *Aspergillus nomius* |
| | | | | HB29S02 | – | |
| | | | | PuliHB29S03 | OM372760 | *Purpureocillium lilacinum* |
| | | | | HB29S04 | – | |
| HB30 | Jingzhou, Hubei | 30.352 N, 112.338 E | Grass | ClroHB3001 | OM372761 | *Clonostachys rosea* |
| | | | | AspsHB3002 | OM372762 | *Aspergillus pseudodeflectus* |
| | | | | HB3003 | – | |
| | | | | AsgrHB3004 | OM372763 | *Aspergillus granulosus* |
| | | | Crop | – | – | |
| HB31 | Jingzhou, Hubei | 30.043 N, 112.158 E | Grass | AsfuHB3101 | OM372764 | *Aspergillus fumigatus* |
| | | | | HB3102 | – | |
| | | | Crop | – | | |
| HN01 | Changsha, Hunan | 28.203 N, 113.303 E | Crop | PulaHN01S01 | OM372765 | *Purpureocillium lavendulum* |
| | | | | AsteHN01S02 | OM372766 | *Aspergillus terreus* |
| | | | | MeanHN01S03 | OM372767 | *Metarhizium anisopliae* |
| | | | Crop | AsfuHN0101 | OM372768 | *Aspergillus fumigatus* |
| | | | | IsjaHN0102 | OM372769 | *Isaria javanica* |
| HN02 | Changde, Hunan | 29.634 N, 111.840 E | Grass | MemaHN0201 | OM372770 | *Metarhizium marquandii* |
| | | | | PulaHN0202 | OM372771 | *Purpureocillium lavendulum* |
| | | | Arbor | PeciHN02S01 | OM372772 | *Penicillium citrinum* |
| | | | | AsteHN02S02 | OM372773 | *Aspergillus terreus* |
| HN03 | Changde, Hunan | 29.131 N, 111.706 E | Grass | HN0301 | – | |
| | | | | PesuHN0302 | OM372774 | *Penicillium subrubescens* |
| | | | Arbor | MemaHN03S01 | OM372775 | *Metarhizium marquandii* |
| | | | | AsnoHN03S02 | OM372776 | *Aspergillus nomius* |
| HN04 | Zhangjiajie, Hunan | 29.424 N, 111.163 E | Orchard | TapiHN04Y01 | OM372777 | Talaromyces pinophilus |
| | | | Grass | – | – | |
| HN05 | Zhangjiajie, Hunan | 29.348 N, 110.568 E | Arbor | HN05S01 | – | |
| | | | Grass | – | – | |
| HN06 | Xiangxi, Hunan | 29.034 N, 110.228 E | Arbor | PulaHN06S01 | OM372778 | *Purpureocillium lavendulum* |
| | | | Grass | TapiHN0601 | OM372779 | Talaromyces pinophilus |
| | | | Crop | HN06Y01 | – | |
| | | | | PuliHN06Y02 | OM372780 | *Purpureocillium lilacinum* |
| | | | | ChasHN06Y03 | OM372781 | *Chloridium aseptatum* |
| | | | | HN06Y04 | – | |
| | | | | HN06Y05 | – | |
| HN07 | Xiangxi, Hunan | 28.623 N, 109.547 E | Grass | HN0701 | – | |
| | | | | Pula sp. HN0702 | OM372782 | *Purpureocillium lavendulum* |
| | | | | PeciHN0703 | OM372783 | *Penicillium citrinum* |
| | | | Arbor | PulaHN07S01 | OM372784 | *Purpureocillium lavendulum* |
| | | | | AsscHN07S02 | OM372785 | *Aspergillus sclerotiorum* |
| HN08 | Huaihua, Hunan | 26.963 N, 109.747 E | Grass | Mema sp. HN0801 | OM372786 | *Metarhizium marquandii* |
| | | | Arbor | PesuHN08S01 | OM372787 | *Penicillium subrubescens* |
| | | | | PulaHN08S02 | OM372788 | *Purpureocillium lavendulum* |
| HN09 | Huaihua, Hunan | 26.614 N, 109.671 E | Grass | HataHN0901 | OM372789 | *Hawksworthiomyces taylorii* |
| | | | | PuliHN0902 | OM372790 | *Purpureocillium lilacinum* |
| | | | | PesuHN0903 | OM372791 | *Penicillium subrubescens* |
| | | | Arbor | PulaHN09S01 | OM372792 | *Purpureocillium lavendulum* |
| | | | | HN09S02 | – | |
| HN10 | Yongzhou, Hunan | 26.662 N, 111.493 E | Grass | HN1001 | – | |
| | | | | PesuHN1002 | OM372793 | *Penicillium subrubescens* |
| | | | | HN1003 | – | |
| | | | Arbor | HN10S01 | – | |
| HN11 | Yongzhou, Hunan | 26.063 N, 111.831 E | Arbor | FufaHN11S01 | OM372794 | *Fusarium falciforme* |
| | | | Fallow land | – | – | |
| HN12 | Yongzhou, Hunan | 25.528 N, 112.111 E | Grass | PulaHN1201 | OM372795 | *Purpureocillium lavendulum* |
| | | | | CudeHN1202 | OM372796 | *Cutaneotrichosporon dermatis* |

**Table A1.** *Cont.*

| | Site | | | Isolate | GenBank Access No. | Species |
|---|---|---|---|---|---|---|
| NO. | Address | Latitude and Longitude | Sample Environment | | | |
| HN13 | Chenzhou, Hunan | 25.659 N, 112.729 E | Grass | TavaHN1301 | OM372797 | *Talaromyces variabilis* |
| | | | | BebaHN1302 | OM372798 | *Beauveria bassiana* |
| | | | | HN1303 | – | |
| | | | Arbor | MeteHN13S01 | OM372799 | *Melanoctona tectonae* |
| HN14 | Chenzhou, Hunan | 25.965 N, 113.042 E | Grass | BebaHN1401 | OM372800 | *Beauveria bassiana* |
| | | | Arbor | PesuHN14S01 | OM372801 | *Penicillium subrubescens* |
| | | | | PuliHN14S02 | OM372802 | *Purpureocillium lilacinum* |
| HN15 | Hengyang, Hunan | 26.426 N, 112.889 E | Grass | AstaHN1501 | OM372803 | *Aspergillus tanneri* |
| | | | | AsscHN1502 | OM372804 | *Aspergillus sclerotiorum* |
| | | | Arbor | AsscHN15S01 | OM372805 | *Aspergillus sclerotiorum* |
| | | | | Mema sp. HN15S02 | OM372806 | *Metarhizium marquandii* |
| | | | | ToalHN15S03 | OM372807 | *Tolypocladium album* |
| | | | | ToalHN15S04 | OM372808 | *Tolypocladium album* |
| | | | | FusoHN15S05 | OM372809 | *Fusarium solani* |
| HN16 | Hengyang, Hunan | 26.974 N, 112.425 E | Grass | Mema sp. HN1601 | OM372810 | *Metarhizium marquandii* |
| | | | Orchard | AsscHN16Z01 | OM372811 | *Aspergillus sclerotiorum* |
| | | | | AsscHN16Z02 | OM372812 | *Aspergillus sclerotiorum* |
| HN17 | Loudi, Hunan | 27.440 N, 112.132 E | Grass | PesuHN1701 | OM372813 | *Penicillium subrubescens* |
| | | | Arbor | MemaHN17S01 | OM372814 | *Metarhizium marquandii* |
| | | | | XepiHN17S02 | OM372815 | *Xenopolyscytalum pinea* |
| | | | | CuelHN17S03 | OM372816 | *Cunninghamella elegans* |
| HN18 | Loudi, Hunan | 27.821 N, 111.763 E | Grass | MemaHN1801 | OM372817 | *Metarhizium marquandii* |
| | | | | HN1802 | | |
| | | | Fallow land | PesuHN18Z01 | OM372818 | *Penicillium subrubescens* |
| HN19 | Yiyang, Hunan | 28.264 N, 111.712 E | Arbor | MeanHN19S01 | OM372819 | *Metarhizium anisopliae* |
| | | | | HN19S02 | | |
| | | | | HN19S03 | | |
| | | | Grass | FufaHN1901 | OM372820 | *Fusarium falciforme* |
| | | | | Mema sp. HN1902 | OM372821 | *Metarhizium marquandii* |
| HN20 | Yiyang, Hunan | 28.525 N, 112.045 E | Grass | BebaHN2001 | OM372822 | *Beauveria bassiana* |
| | | | | PulaHN2002 | OM372823 | *Purpureocillium lavendulum* |
| | | | | PesuHN2003 | OM372824 | *Penicillium subrubescens* |
| | | | Fallow land | HN20Z01 | – | |
| | | | | BebaHN20Z02 | OM372825 | *Beauveria bassiana* |
| | | | | Mema sp. HN20Z03 | OM372826 | *Metarhizium marquandii* |
| HN21 | Changsha, Hunan | 28.222 N, 112.567 E | Crop | MeanHN21G01 | OM372829 | *Metarhizium anisopliae* |
| | | | | Pula sp. HN21G02 | OM372830 | *Purpureocillium lavendulum* |
| | | | | TapiHN21G03 | OM372831 | *Talaromyces pinophilus* |
| | | | Arbor | PulaHN21S01 | OM372827 | *Purpureocillium lavendulum* |
| | | | | Mema sp. HN21S02 | OM372828 | *Metarhizium marquandii* |
| HN22 | Xiangtan, Hunan | 27.806 N, 112.511 E | Fallow land | Pula sp. HN22Z01 | OM372832 | *Purpureocillium lavendulum* |
| | | | Arbor | – | – | |
| HN23 | Xiangtan, Hunan | 27.846 N, 113.017 E | Grass | Mema sp. HN2301 | OM372833 | *Metarhizium marquandii* |
| | | | | PulaHN2302 | OM372834 | *Purpureocillium lavendulum* |
| | | | | PeciHN2303 | OM372835 | *Penicillium citrinum* |
| | | | Arbor | Pula sp. HN23S01 | OM372836 | *Purpureocillium lavendulum* |
| HN24 | Hengyang, Hunan | 27.229 N, 112.897 E | Grass | TavaHN2401 | OM372837 | *Talaromyces variabilis* |
| | | | | ApcaHN2402 | OM372838 | *Apiotrichum cacaoliposimilis* |
| | | | | PesuHN2403 | OM372839 | *Penicillium subrubescens* |
| | | | Arbor | MeanHN24S01 | OM372840 | *Metarhizium anisopliae* |
| | | | | HN24S02 | – | |
| HN25 | Zhuzhou, Hunan | 26.893 N, 113.374 E | Grass | Mema sp. HN2501 | OM372841 | *Metarhizium marquandii* |
| | | | Orchard | MeanHN25B01 | OM372842 | *Metarhizium anisopliae* |

**Table A1.** *Cont.*

| | Site | | | Isolate | GenBank Access No. | Species |
|---|---|---|---|---|---|---|
| NO. | Address | Latitude and Longitude | Sample Environment | | | |
| HN26 | Zhuzhou, Hunan | 27.496 N, 113.486 E | Arbor | MemaHN26S01 HN26S02 | OM372843 – | *Metarhizium marquandii* |
| | | | Fallow land | MeanHN26Z01 | OM372844 | *Metarhizium anisopliae* |
| HN27 | Xiangxi, Hunan | 27.914 N, 109.385 E | Grass | PulaHN2701 | OM372845 | *Purpureocillium lavendulum* |
| | | | | PhliHN2702 | OM372846 | *Phialophora livistonae* |
| HN28 | Huaihua, Hunan | 27.896 N, 109.702 E | Orchard | HN28J01 | – | |
| | | | | PuliHN28J02 | OM372847 | *Purpureocillium lilacinum* |
| | | | | PeruHN28J03 | OM372848 | *Penicillium rubens* |
| HN29 | Huaihua, Hunan | 27.367 N, 109.935 E | Fallow land | PulaHN29B01 | OM372849 | *Purpureocillium lavendulum* |
| | | | | MeanHN29B02 | OM372850 | *Metarhizium anisopliae* |
| | | | Grass | ArhiHN2901 | OM372851 | *Arthropsis hispanica* |
| HN30 | Huaihua, Hunan | 27.216 N, 110.420 E | Arbor | SimiHN3001 | OM372852 | *Simplicillium minatense* |
| | | | | IsjaHN3002 | OM372853 | *Isaria javanica* |
| HN31 | Shaoyang, Hunan | 26.941 N, 110.638 E | Arbor | XepiHN3101 | OM372854 | *Xenopolyscytalum pinea* |
| HN32 | Shaoyang, Hunan | 26.322 N, 110.837 E | Grass | – | – | |
| HE01 | Xingtai, Hebei | 36.905 N, 114.559 E | Crop | MeanHE01A01 | OM372855 | *Metarhizium anisopliae* |
| | | | | LecoHE01A02 | OM372856 | *Lecanicillium coprophilum* |
| | | | Grass | MeanHE01B01 | OM372857 | *Metarhizium anisopliae* |
| | | | | NemaHE01B02 | OM372858 | *Nectria mauritiicola* |
| | | | | LecoHE01B03 | OM372859 | *Lecanicillium coprophilum* |
| HE02 | Shijiazhuang, Hebei | 35.994 N, 113.758 E | Arbor | LecoHE02A01 | OM372860 | *Lecanicillium coprophilum* |
| | | | | MeanHE02A02 HE02A03 | OM372861 – | *Metarhizium anisopliae* |
| HE03 | Baoding, Hebei | 39.138 N, 115.536 E | Crop | LecoHE03A01 | OM372862 | *Lecanicillium coprophilum* |
| | | | | MemaHE03A02 | OM372863 | *Metarhizium marquandii* |
| | | | Grass | HE03B01 | – | |
| | | | | LecoHE03B02 | OM372864 | *Lecanicillium coprophilum* |
| | | | Poplar | ClgrHE03C01 | OM372865 | *Clonostachys grammicospora* |
| | | | | MeanHE03C02 | OM372866 | *Metarhizium anisopliae* |
| | | | | TatrHE03C03 | OM372867 | *Talaromyces trachyspermus* |
| HE04 | Zhangjiakou, Hebei | 39.273 N, 115.455 E | Poplar | AualHE04A01 | OM372868 | *Auxarthron alboluteum* |
| | | | | ClgrHE04A02 | OM372869 | *Clonostachys grammicospora* |
| | | | Crop | HE04B01 | – | |
| | | | | LecoHE04B02 | OM372870 | *Lecanicillium coprophilum* |
| | | | | Mema sp. HE04B03 | OM372871 | *Metarhizium marquandii* |
| HE05 | Zhangjiakou, Hebei | 39.375 N, 114.866 E | Poplar | HE05A01 | – | |
| | | | | LecoHE05A02 | OM372872 | *Lecanicillium coprophilum* |
| | | | Crop | Mema sp. HE05B01 | OM372873 | *Metarhizium marquandii* |
| | | | | MeanHE05B02 | OM372874 | *Metarhizium anisopliae* |
| HE06 | Zhangjiakou, Hebei | 40.488 N, 114.838 E | Orchard | MeanHE06A01 | OM372875 | *Metarhizium anisopliae* |
| | | | | TrteHE06A02 | OM372876 | *Trichurus terrophilus* |
| | | | | AssyHE06A03 | OM372877 | *Aspergillus sydowii* |
| | | | | BebaHE06A04 | OM372878 | *Beauveria bassiana* |
| | | | Crop | Mema sp. HE06B01 | OM372879 | *Metarhizium marquandii* |
| | | | | PuliHE06B02 | OM372880 | *Purpureocillium lilacinum* |
| HE07 | Zhangjiakou, Hebei | 41.267 N, 114.785 E | Crop | HE07A01 | – | |
| | | | | AscrHE07C01 | OM372882 | *Aspergillus crustosus* |
| | | | Grass | LecoHE07B01 | OM372881 | *Lecanicillium coprophilum* |
| | | | Poplar | LecoHE07D01 | OM372883 | *Lecanicillium coprophilum* |
| HE08 | Zhangjiakou, Hebei | 41.073 N, 115.389 E | Grass | HE08A01 HE08A02 | – – | |
| | | | Crop | AualHE08B01 | OM372884 | *Auxarthron alboluteum* |
| | | | | AsfuHE08C01 | OM372885 | *Aspergillus fumigatus* |
| HE09 | Chengde, Hebei | 41.581 N, 116.023 E | Grass | MeanHE09A01 | OM372886 | *Metarhizium anisopliae* |
| | | | | PulaHE09A02 | OM372887 | *Purpureocillium lavendulum* |
| | | | Elm | AsfuHE09B01 | OM372888 | *Aspergillus fumigatus* |
| | | | Crop | HE09C01 | – | |
| | | | | OifuHE09C02 | OM372889 | *Oidiodendron fuscum* |
| HE10 | Chengde, Hebei | 42.001 N, 116.975 E | Grass | TapuHE10A01 | OM372890 | *Talaromyces purpureogenus* |
| | | | Crop | – | – | |

**Table A1.** *Cont.*

| | Site | | | Isolate | GenBank Access No. | Species |
|---|---|---|---|---|---|---|
| NO. | Address | Latitude and Longitude | Sample Environment | | | |
| HE11 | Chengde, Hebei | 42.253 N, 117.143 E | Grass | PulaHE11A01 | OM372891 | *Purpureocillium lavendulum* |
| | | | | PesuHE11A02 | OM372892 | *Penicillium subrubescens* |
| | | | Pine | CofaHE11C01 | OM372893 | *Coniochaeta fasciculata* |
| | | | Crop | AssyHE11D01 | OM372894 | *Aspergillus sydowii* |
| HE12 | Chengde, Hebei | 41.997 N, 117.655 E | Orchard | AsfuHE12A01 | OM372895 | *Aspergillus fumigatus* |
| | | | Grass | PulaHE12B01 | OM372896 | *Purpureocillium lavendulum* |
| | | | | MeanHE12B02 | OM372897 | *Metarhizium anisopliae* |
| HE13 | Chengde, Hebei | 41.302 N, 118.038 E | Crop | HE13A01 | – | |
| | | | | MeanHE13B01 | OM372898 | *Metarhizium anisopliae* |
| | | | | MeanHE13B02 | OM372899 | *Metarhizium anisopliae* |
| | | | Poplar | AsudHE13C01 | OM372900 | *Aspergillus udagawae* |
| | | | | PulaHE13C02 | OM372901 | *Purpureocillium lavendulum* |
| | | | | MeanHE13C03 | OM372902 | *Metarhizium anisopliae* |
| HE14 | Chengde, Hebei | 40.578 N, 117.704 E | Crop | TatrHE14A01 | OM372903 | *Talaromyces trachyspermus* |
| | | | Grass | – | – | |
| HE15 | Tangshan, Hebei | 40.108 N, 117.985 E | Crop | MeanHE15A01 | OM372904 | *Metarhizium anisopliae* |
| | | | | Sicy sp. HE15A02 | OM372905 | *Simplicillium cylindrosporum* |
| | | | Arbor | MeanHE15B01 | OM372906 | *Metarhizium anisopliae* |
| | | | | PemiHB15B02 | OM372907 | *Penicillium mirabile* |
| | | | Grass | SicyHE15C01 | OM372908 | *Simplicillium cylindrosporum* |
| | | | | SimiHE15C02 | OM372909 | *Simplicillium minatense* |
| | | | | MeanHE15C03 | OM372910 | *Metarhizium anisopliae* |
| HE16 | Tangshan, Hebei | 39.584 N, 118.264 E | Grass | MeanHE16A01 | OM372911 | *Metarhizium anisopliae* |
| HE17 | Tangshan, Hebei | 39.490 N, 118.682 E | Grass | PeciHE17A01 | OM372912 | *Penicillium citrinum* |
| | | | | SicyHE17A02 | OM372913 | *Simplicillium cylindrosporum* |
| | | | | PulaHE17A03 | OM372914 | *Purpureocillium lavendulum* |
| | | | Orchard | Pula sp. HE17B01 | OM372915 | *Purpureocillium lavendulum* |
| | | | | SicyHE17B02 | OM372916 | *Simplicillium cylindrosporum* |
| | | | | BebaHE17B03 | OM372917 | *Beauveria bassiana* |
| | | | Poplar | SimiHE17C01 | OM372918 | *Simplicillium minatense* |
| | | | | MemaHE17C02 | OM372919 | *Metarhizium marquandii* |
| HE18 | Tangshan, Hebei | 39.408 N, 117.954 E | Crop | PulaHE18A01 | OM372920 | *Purpureocillium lavendulum* |
| | | | | TatrHE18B01 | OM372921 | *Talaromyces trachyspermus* |
| | | | | Pula sp. HE18B02 | OM372922 | *Purpureocillium lavendulum* |
| | | | | TrteHE18B03 | OM372923 | *Trichurus terrophilus* |
| | | | Poplar | MeanHE18C01 | OM372924 | *Metarhizium anisopliae* |
| HE19 | Tianjin, Hebei | 38.768 N, 117.184 E | Crop | MeanHE19A01 | OM372925 | *Metarhizium anisopliae* |
| | | | | NemaHE19A02 | OM372926 | *Nectria mauritiicola* |
| | | | Orchard | Pula sp. HE19B01 | OM372927 | *Purpureocillium lavendulum* |
| | | | | MeanHE19B02 | OM372928 | *Metarhizium anisopliae* |
| HE20 | Cangzhou, Hebei | 38.151 N, 115.740 E | Crop | PulaHE20A01 | OM372929 | *Purpureocillium lavendulum* |
| | | | | MemaHE20A02 | OM372930 | *Metarhizium marquandii* |
| | | | Grass | MeanHE20B01 | OM372931 | *Metarhizium anisopliae* |
| HE21 | Hengshui, Hebei | 37.719 N, 115.193 E | Crop | PemiHB21A01 | OM372932 | *Penicillium mirabile* |
| | | | Grass | PechHE21B01 | OM372933 | *Penicillium chrysogenum* |
| | | | | Pula sp. HE21B02 | OM372934 | *Purpureocillium lavendulum* |
| | | | | MemaHE21B03 | OM372935 | *Metarhizium marquandii* |
| HE22 | Handan, Hebei | 36.804 N, 115.193 E | Crop | ClroHE22A01 | OM372936 | *Clonostachys rosea* |
| | | | | MemaHe22B01 | OM372937 | *Metarhizium marquandii* |
| HA01 | Xinxiang, Henan | 35.268 N, 113.974 E | Orchard | HA01A01 | – | |
| | | | Crop | MemaHA01B01 | OM372938 | *Metarhizium marquandii* |
| | | | Grass | – | – | |
| HA02 | Linzhou, Henan | 35.994 N, 113.758 N | Crop | PulaHA02A01 | OM372939 | *Purpureocillium lavendulum* |
| | | | | MeanHA02B01 | OM372940 | *Metarhizium anisopliae* |
| | | | Arbor | PulaHA02C01 | OM372941 | *Purpureocillium lavendulum* |
| HA03 | Linzhou, Henan | 35.928 N, 113.655 E | Crop | – | – | |
| HA04 | Puyang, Henan | 36.090 N, 115.124 E | Crop | TrteHA04A01 | OM372942 | *Trichurus terrophilus* |
| | | | | MemaHA04B01 | OM372943 | *Metarhizium marquandii* |
| | | | | HA04B02 | – | |
| | | | Grass | Pula sp. HA04C01 | OM372944 | *Purpureocillium lavendulum* |

**Table A1.** *Cont.*

| | Site | | | Isolate | GenBank Access No. | Species |
|---|---|---|---|---|---|---|
| NO. | Address | Latitude and Longitude | Sample Environment | | | |
| HA05 | Kaifeng, Henan | 34.790 N, 114.485 E | Crop | – | – | |
| | | | Grass | PulaHA05B01 | OM372945 | *Purpureocillium lavendulum* |
| | | | | BebaHA05B02 | OM372946 | *Beauveria bassiana* |
| | | | Crop | Mema sp. HA05C01 | OM372947 | *Metarhizium marquandii* |
| | | | | TavaHA05C02 | OM372948 | *Talaromyces variabilis* |
| HA06 | Kaifeng, Henan | 34.895 N, 114.328 E | Crop | MemaHA06A01 | OM372949 | *Metarhizium marquandii* |
| | | | | PulaHA06A02 | OM372950 | *Purpureocillium lavendulum* |
| | | | | HA06A03 | – | |
| | | | Poplar | OifuHA06B01 | OM372951 | *Oidiodendron fuscum* |
| | | | | HA06B02 | – | |
| | | | Grass | PulaHA06C01 | OM372952 | *Purpureocillium lavendulum* |
| | | | | ChloHA06C02 | OM372953 | *Chrysosporium lobatum* |
| | | | | MeanHA06C03 | OM372954 | *Metarhizium anisopliae* |
| HA07 | Zhengzhou, Henan | 34.481 N, 113.030 E | Arbor | MemaHA07A01 | OM372955 | *Metarhizium marquandii* |
| | | | Orchard | PeciHA07B01 | OM372956 | *Penicillium citrinum* |
| | | | | Pula sp. HA07B02 | OM372957 | *Purpureocillium lavendulum* |
| | | | Crop | PeguHA07C01 | OM372958 | *Penicillium guttulosum* |
| HA08 | Luoyang, Henan | 34.555 N, 112.873 E | Crop | PulaHA08A01 | OM372959 | *Purpureocillium lavendulum* |
| | | | | MemaHA08A02 | OM372960 | *Metarhizium marquandii* |
| | | | | PulaHA08B01 | OM372961 | *Purpureocillium lavendulum* |
| | | | | MeanHA08B02 | OM372962 | *Metarhizium anisopliae* |
| | | | | MemaHA08B03 | OM372963 | *Metarhizium marquandii* |
| | | | | PulaHA08C01 | OM372964 | *Purpureocillium lavendulum* |
| | | | | MemaHA08C02 | OM372965 | *Metarhizium marquandii* |
| HA09 | Luoyang, Henan | 34.768 N, 112.093 E | Crop | PulaHA09A01 | OM372966 | *Purpureocillium lavendulum* |
| | | | | MeanHA09A02 | OM372967 | *Metarhizium anisopliae* |
| | | | Poplar | MemaHA09B01 | OM372968 | *Metarhizium marquandii* |
| | | | | MeanHA09B02 | OM372969 | *Metarhizium anisopliae* |
| | | | | PulaHA09B03 | OM372970 | *Purpureocillium lavendulum* |
| HA10 | Sanmenxia, Henan | 34.797 N, 111.243 E | Crop | PulaHA10A01 | OM372971 | *Purpureocillium lavendulum* |
| | | | | MeanHA10A02 | OM372972 | *Metarhizium anisopliae* |
| | | | | Pula sp. HA10B01 | OM372973 | *Purpureocillium lavendulum* |
| | | | | MemaHA10B02 | OM372974 | *Metarhizium marquandii* |
| | | | | MeanHA10B03 | OM372975 | *Metarhizium anisopliae* |
| HA11 | Sanmenxia, Henan | 34.626 N, 110.914 E | Crop | MemaHA11A01 | OM372976 | *Metarhizium marquandii* |
| | | | | MeanHA11A02 | OM372977 | *Metarhizium anisopliae* |
| | | | | PulaHA11A03 | OM372978 | *Purpureocillium lavendulum* |
| | | | | PulaHA11B01 | OM372979 | *Purpureocillium lavendulum* |
| | | | Grass | – | – | |
| HA12 | Nanyang, Henan | 33.566 N, 111.185 E | Crop | Pula sp. HA12A01 | OM372980 | *Purpureocillium lavendulum* |
| | | | | Mema sp. HA12A02 | OM372981 | *Metarhizium marquandii* |
| | | | | BebaHA12B01 | OM372982 | *Beauveria bassiana* |
| | | | | PuliHA12B02 | OM372983 | *Purpureocillium lilacinum* |
| | | | | TrteHA12B03 | OM372984 | *Trichurus terrophilus* |
| | | | Grass | PulaHA12C01 | OM372985 | *Purpureocillium lavendulum* |
| HA13 | Nanyang, Henan | 33.072 N, 111.792 E | Crop | – | – | – |
| | | | | PuliHA13B01 | OM372986 | *Purpureocillium lilacinum* |
| | | | | HA13B02 | – | |
| HA14 | Nanyang, Henan | 32.780 N, 112.707 E | Crop | PulaHA14A01 | OM372987 | *Purpureocillium lavendulum* |
| | | | | NemaHA14B01 | OM372988 | *Nectria mauritiicola* |
| | | | Grass | MemaHA14C01 | OM372989 | *Metarhizium marquandii* |
| HA15 | Xinyang, Henan | 32.401 N, 113.931 E | Crop | PulaHA15A01 | OM372990 | *Purpureocillium lavendulum* |
| | | | | ChasHA15A02 | OM372991 | *Chloridium aseptatum* |
| | | | Grass | PulaHA15C01 | OM372992 | *Purpureocillium lavendulum* |
| HA16 | Xinyang, Henan | 32.338 N, 114.128 E | Crop | PulaHA16A01 | OM372993 | *Purpureocillium lavendulum* |
| | | | | MaauHA16A02 | OM372994 | *Malbranchea aurantiaca* |
| | | | Grass | PulaHA16C01 | OM372995 | *Purpureocillium lavendulum* |
| HA17 | Zhumadian, Henan | 32.707 N, 114.109 E | Crop | PulaHA17A01 | OM372996 | *Purpureocillium lavendulum* |
| | | | | MemaHA17B01 | OM372997 | *Metarhizium marquandii* |
| | | | | PulaHA17B02 | OM372998 | *Purpureocillium lavendulum* |
| | | | Grass | HA17C01 | – | – |
| | | | | PulaHA17C02 | OM372999 | *Purpureocillium lavendulum* |

**Table A1.** *Cont.*

| | Site | | | Isolate | GenBank Access No. | Species |
|---|---|---|---|---|---|---|
| NO. | Address | Latitude and Longitude | Sample Environment | | | |
| HA18 | Luohe, Henan | 33.510 N, 113.980 E | Crop | PulaHA18A01 | OM373000 | *Purpureocillium lavendulum* |
| | | | | Pula sp. HA18B01 | OM373001 | *Purpureocillium lavendulum* |
| | | | | HA18B02 | – | |
| | | | Grass | MeanHA18C01 | OM373002 | *Metarhizium anisopliae* |
| | | | | PulaHA18C02 | OM373003 | *Purpureocillium lavendulum* |
| | | | | MemaHA18C03 | OM373004 | *Metarhizium marquandii* |
| HA19 | Pingdingshan, Henan | 33.652 N, 113.370 E | Crop | MeanHA19A01 | OM373005 | *Metarhizium anisopliae* |
| | | | | PulaHA19A02 | OM373006 | *Purpureocillium lavendulum* |
| | | | | MeanHA19B01 | OM373007 | *Metarhizium anisopliae* |
| | | | | PulaHA19B02 | OM373008 | *Purpureocillium lavendulum* |
| | | | Grass | MemaHA19C01 | OM373009 | *Metarhizium marquandii* |
| | | | | Pula sp. HA19C02 | OM373010 | *Purpureocillium lavendulum* |
| | | | | AssyHA19C03 | OM373011 | *Aspergillus sydowii* |
| HA20 | Xuchang, Henan | 34.052 N, 113.709 E | Crop | MemaHA20A01 | OM373012 | *Metarhizium marquandii* |
| | | | | MemaHA20B01 | OM373013 | *Metarhizium marquandii* |
| | | | Grass | PulaHA20C01 | OM373014 | *Purpureocillium lavendulum* |
| | | | | MemaHA20C02 | OM373015 | *Metarhizium marquandii* |
| | | | | BebaHA20C03 | OM373016 | *Beauveria bassiana* |
| HA21 | Zhoukou, Henan | 33.978 N, 114.867 E | Crop | PulaHA21A01 | OM373017 | *Purpureocillium lavendulum* |
| | | | | MemaHA2102 | OM373018 | *Metarhizium marquandii* |
| | | | Grass | Pula sp. HA21B01 | OM373019 | *Purpureocillium lavendulum* |
| | | | | PeciHA21B02 | OM373020 | *Penicillium citrinum* |
| HA22 | Shangqiu, Henan | 34.350 N, 115.572 E | Crop | HA22A01 | – | |
| | | | | BebaHA22A02 | OM373021 | *Beauveria bassiana* |
| | | | | PuliHA22B01 | OM373022 | *Purpureocillium lilacinum* |
| | | | Grass | PuliHA22C01 | OM373023 | *Purpureocillium lilacinum* |
| HA23 | Shangqiu, Henan | 34.596 N, 115.109 E | Crop | TapuHA23A01 | OM373024 | *Talaromyces purpureogenus* |
| | | | | MemaHA23B01 | OM373025 | *Metarhizium marquandii* |
| | | | | HA23B02 | – | |
| | | | Orchard | PemiHA23C01 | OM373026 | *Penicillium mirabile* |
| HA24 | Kaifeng, Henan | 34.771 N, 114.806 E | Crop | MemaHA24A01 | OM373027 | *Metarhizium marquandii* |
| | | | | PulaHA24A02 | OM373028 | *Purpureocillium lavendulum* |
| | | | | PuliHA24B01 | OM373029 | *Purpureocillium lilacinum* |
| HA25 | Zhengzhou, Henan | 34.838 N, 114.036 E | Crop | PeciHA25A01 | OM373030 | *Penicillium citrinum* |
| | | | | MeanHA25A02 | OM373031 | *Metarhizium anisopliae* |
| | | | Grass | MeanHA25B01 | OM373032 | *Metarhizium anisopliae* |
| | | | | PuliHA25B02 | OM373033 | *Purpureocillium lilacinum* |
| | | | Crop | MeanHA25C01 | OM373034 | *Metarhizium anisopliae* |
| | | | Poplar | MemaHA25D01 | OM373035 | *Metarhizium marquandii* |
| | | | | HA25D02 | – | |

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
