# Peer review of "Entomopathogenic Fungi in the Soils of China and Their Bioactivity against Striped Flea Beetles Phyllotretastriolata"

_diversity, doi:10.3390/d14060464_

Round 1
Reviewer 1 Report
I made several comments and corrections however I didn't read the whole manuscript. First I recommended authors extensive editing of the English language and the style required improvement.

Author Response
Dear Editor and Reviewer:
Thank you for taking time out of your busy schedule to review the manuscript.
I would like to thank you for suggestions on the manuscript. Based on the comments of reviewers, I have revised all possible errors highlighted in the article. Thank you again for your careful modification.
To sum up, we have carefully revised our manuscript. Please do not hesitate to contact us if you have any question. Thanks again for your hard work. Best wishes to you!
Author: Ke Zhang
Date: 14 May 2022
Reviewer 2 Report
The topic could be of some interest because wants to investigate the entomopathogenic fungal diversity in soils of four Chinese regions whose findings could have some application to naturally control pests.
Anyway, the manuscript suffers from several flaws (e.g. the balance between the “diversity” and “applicative” study is missing) others are methodological (Isolation, the molecular marker used for fungal identification, the phylogeny claimed but not done/reliable etc..), the result section reports data not well introduced in M&M; frequently the sentences sound confusing and the English should be carefully revised. For these reasons, the publication of this manuscript is strongly discouraged. In consideration of the changes needed and the time necessary to address all concerns raised, I reject the manuscript to give time to the author to work on it.
Below are some additional not exhaustive notes.
The abstract should be rewritten following the basic rules of the scientific report. Now is a sequence of puzzling sentences. The topic is “Entomopathogenic Fungi”, so it cannot start talking about Chinese provinces. Aspergillus is a genus accounting for more than 300 species, could you please be more precise?
Keywords. The majority of KW chosen by authors are already present in the title and then are not fruitful in improving the article finding. Here you can find some suggestions for choosing KW: https://falconediting.com/en/blog/6-tips-for-choosing-keywords-for-your-scientific-manuscript
https://blog.wordvice.com/choosing-research-paper-keywords/
Introduction
L22-23 “Entomopathogenic fungi (EPF) are the first microorganisms found to cause insect diseases, and also the first microorganisms used for biological control” 1) are you sure? What about Bacillus thuringensis? 2) missing citation
L25-26 In addition to providing nutrients for each other,s to growth, some pathogenic microorganisms, especially EPF, can also increase pest population and control populations at low levels for long periods. The sentence is not clear, please rework it.
L37 remove etc
L39 Many countries in the world have done a lot of work in the research and utilization 39 of EPF. Add reference
L52-53 is an important pest of cruciferous, solanaceae, melons and legumes vegetables. The sentence is not consistent because are cited cruciferous, an adjective (of a gone family, now Brassicaceae), a family (it should be capitalized), and a common name (e.g. melon)
53-54 The management is based on synthetic chemical insecticides [16], leading to insecticide resistance [17]. Word repetition (insecticide/s); comma not necessary; citations could be grouped at the end of the sentence and punctuation cannot be placed to allow a close citation.
L54-58 please rework, these sentences are confusing.
L62-63 B. bassiana and M. anisopliae.. P. striolata the species names should be in Italics
L64-65 please rework the sentence is confusing
L71-72 However, the distributions of 71 soil EPF in these regions are not clear. Do you mean that has not been investigated yet?
L75 to explore, between them there is an exceeding space.
L79 fromsites missing space, rework the sentence.
L84 wascollected, mixedand missing spaces, rework the sentence, it’s confusing.
L85 collected in China, exceeding information, you already tell that sampling has been performed on Chinese provinces.
Figure 1 is unreadable. If the authors want to highlight the sampling points no other cities, rivers and so on should be indicated except the region used for sampling. Considering the huge data reported and the importance of the sampling point in the research design, I suggest moving it to supplementary material.
L93 The method of our previous work was used to [27] isolate. Citations should be reported just before the closest punctuation mark; the used method is described below and so the citation is not necessary. Removing plant debris and stone is not a pre-treatment.
L95-96 selective medium (PDA, 0.2 g/L cycloheximide, 0.2 95 g/L chloramphenicol and 0.0133 g/L rose Bengal sodium). The work purpose is to isolate entomopathogenic fungi, if you tell me you used a selective medium I expect to find chitin in the recipe. The culture medium you used favours fungi because inhibits bacteria, Rose Bengal slow-down the fast-growing fungi, but with 20g/l of dextrose the main part of ascomycetes can grow very well. Moreover, the temperature of incubation is missing; a parameter that could make a difference in isolation.
Identification of fungal species and analysis of genetic homology
- It is well known that ITS is not useful for species-level identification in large groups such as Penicillium, Aspergillus, Fusarium, Trichoderma, Cladosporium and others
- Some species are cryptic and assessed by molecular methods only.
In this light, part of the identifications performed are not reliable.
L99 The method of our previous work [27] was referred. As before, the methods should be reported for reproducibility etc (see above)
L104 The PCR products were visualized with UV illumination photographed (Tanon-1600, Tanon.. This step is not important, please discard it. It would more useful the program you used to amplify ITS
L107-108 obtained ITS - rDNA sequences were submmited to Gen-107 Bank and compared with similar sequences through the BLAST of NCBI. What conditions did you use? I mean did you add some constraint?
L108-109 The phylogenetic trees of EPF were constructed by MEGA X [27] with a statistical method of maximum likelihood, bootstrap test of 500 replications and the Jukes-Cantor model. This sounds as a simple clustering and not as a phylogenetic analysis. In any case, to get clustering reliability more reference strains should be reported, I mean not only those for which EP trait has been reported.
L130 “The samples”- what kind of samples, what the authors want to assess?
L136 Analysis of the data was done . what kind of data? I suppose those of the bioassay. Move this section before SEM I
Results
EPF species diversity in soils of Hebei and Central China
Why separate Hebei from other regions when the materials and methods have been reported as four regions. This section is quite confusing and cannot be reported as a list of species. All the results achieved should be introduced in the materials and methods so the groupings (regional/ crops vs grass vs follow lands) should be claimed before the aim of the study could be clear.
What result do you expect comparing fungi isolated from different regions? what do you expect comparing the outcomes from different soils…please explain
302 pure fungal strains from 226 soil samples sound like a poor yield. I wonder if you performed a selection of the isolates. Please explain
L190-192 Aspergillus, Fusarium, Lecanicillium, 189 Metarhizium, Monascus and Talaromyces all had obvious pathogenicityto P. striolata (Table 190 4). Besides, we found that Aspergillus, Lecanicillium, Monascus and Talaromyces had the pathogenicity. Genus and species cannot be used with the same meaning, especially when the general trait of a genus is not associated with the EP trait.
Discussion and conclusions are affected by the abovementioned weakness, so should be reworked
Minor notes
Bioactivity in the title mustn’t be trunked
References from 59 to 64 are red
L457 Claviceps should be capitalised
Author Response
Dear Editor and Reviewer:
Thank you for taking time out of your busy schedule to review the manuscript. Now we have carefully correted and replied the manuscript for this revision. The explanation is as follows:
The abstract should be rewritten following the basic rules of the scientific report. Now is a sequence of puzzling sentences. The topic is “Entomopathogenic Fungi”, so it cannot start talking about Chinese provinces. Aspergillus is a genus accounting for more than 300 species, could you please be more precise?
A: Delete the first sentence and rewrite, Aspergillus spp.refer to several aspergillus species in table 4.
L22-23 “Entomopathogenic fungi (EPF) are the first microorganisms found to cause insect diseases, and also the first microorganisms used for biological control” 1) are you sure? What about Bacillus thuringensis? 2) missing citation
A: This statement is wrong. Entomopathogenic fungi (EPF) are ubiquitous in nature, the biological plant protection with EPF has a key role in a sustainable pest management program.
L25-26 In addition to providing nutrients for each other,s to growth, some pathogenic microorganisms, especially EPF, can also increase pest population and control populations at low levels for long periods. The sentence is not clear, please rework it.
A: In addition to absorbing nutrients for growth, some pathogenic microorganisms, especially EPF, can also control insect populations at low levels for long periods.
L37 remove etc
A: OK, thank you for your advice.
L39 Many countries in the world have done a lot of work in the research and utilization 39 of EPF. Add reference
A: This sentence is our own summary, can be deleted.
L52-53 is an important pest of cruciferous, solanaceae, melons and legumes vegetables. The sentence is not consistent because are cited cruciferous, an adjective (of a gone family, now Brassicaceae), a family (it should be capitalized), and a common name (e.g. melon)
A: Phyllotreta striolata (Coleoptera: Chrysomelidae) is an important pest of Brassicaceae, Solanaceae, Cucurbitaceae and Leguminosae vegetable.
53-54 The management is based on synthetic chemical insecticides [16], leading to insecticide resistance [17]. Word repetition (insecticide/s); comma not necessary; citations could be grouped at the end of the sentence and punctuation cannot be placed to allow a close citation.
L54-58 please rework, these sentences are confusing.
A: OK, The management is based on synthetic chemical pesticide leading to insect resistance [17, 18]. Few registered varieties of biopesticides, which cannot meet the needs of green prevention and control. EPF represent the most promising candidate in the Integrated Pest Management (IPM) program approach [19].
L62-63 B. bassiana and M. anisopliae.. P. striolata the species names should be in Italics
A: OK, thank you for your advice.
L64-65 please rework the sentence is confusing
A: It was found that B. bassiana and M. anisopliae could infect the larvae and adults of P. striolata [22, 23], but the research was still at the laboratory level. In view of most EPF are soil-dwelling microbes, thus, investigating soil fungi will be benefit to exploring new species of EPF resources [24-26].
L71-72 However, the distributions of 71 soil EPF in these regions are not clear. Do you mean that has not been investigated yet?
A: No related reports have been seen.
L75 to explore, between them there is an exceeding space.
A: OK, thank you for your advice.
L79 fromsites missing space, rework the sentence.
A: OK, thank you for your advice.
L84 wascollected, mixedand missing spaces, rework the sentence, it’s confusing.
A: OK, thank you for your advice.
L85 collected in China, exceeding information, you already tell that sampling has been performed on Chinese provinces.
A: OK, thank you for your advice.
Figure 1 is unreadable. If the authors want to highlight the sampling points no other cities, rivers and so on should be indicated except the region used for sampling. Considering the huge data reported and the importance of the sampling point in the research design, I suggest moving it to supplementary material.
A: The map is to show the sampling sites. Attached is a detailed sampling site, the map can be deleted.
L93 The method of our previous work was used to [27] isolate. Citations should be reported just before the closest punctuation mark; the used method is described below and so the citation is not necessary. Removing plant debris and stone is not a pre-treatment.
A: OK, thank you for your advice.
L95-96 selective medium (PDA, 0.2 g/L cycloheximide, 0.2 95 g/L chloramphenicol and 0.0133 g/L rose Bengal sodium). The work purpose is to isolate entomopathogenic fungi, if you tell me you used a selective medium I expect to find chitin in the recipe. The culture medium you used favours fungi because inhibits bacteria, Rose Bengal slow-down the fast-growing fungi, but with 20g/l of dextrose the main part of ascomycetes can grow very well. Moreover, the temperature of incubation is missing; a parameter that could make a difference in isolation.
A: Yes, Bengal red selective media can suppress the size of mold colonies that multiply too quickly, so that other fungi can be screened. This part of the description is not complete and has been revised.
Identification of fungal species and analysis of genetic homology
- It is well known that ITS is not useful for species-level identification in large groups such as Penicillium, Aspergillus, Fusarium, Trichoderma, Cladosporium and others
- Some species are cryptic and assessed by molecular methods only.
In this light, part of the identifications performed are not reliable.
A: We first compared the nucleic acid sequences of the sequenced strains in the NCBI database to determine their possible species, and then constructed a phylogenetic tree of the strain and a large number of reported strains of the same genus. Finally, the strains of the same genus were put together to construct a phylogenetic tree again.
L99 The method of our previous work [27] was referred. As before, the methods should be reported for reproducibility etc (see above)
A: OK, thank you for your advice.
L104 The PCR products were visualized with UV illumination photographed (Tanon-1600, Tanon.. This step is not important, please discard it. It would more useful the program you used to amplify ITS
A: OK, thank you for your advice.
L107-108 obtained ITS - rDNA sequences were submmited to Gen-107 Bank and compared with similar sequences through the BLAST of NCBI. What conditions did you use? I mean did you add some constraint?
A: program selection : Highly similar sequences (megablast). The comparison was only to determine the species roughly, and the species was determined by comprehensive phylogenetic tree analysis.
L108-109 The phylogenetic trees of EPF were constructed by MEGA X [27] with a statistical method of maximum likelihood, bootstrap test of 500 replications and the Jukes-Cantor model. This sounds as a simple clustering and not as a phylogenetic analysis. In any case, to get clustering reliability more reference strains should be reported, I mean not only those for which EP trait has been reported.
A: We first compared the nucleic acid sequences of the sequenced strains in the NCBI database to determine their possible species, and then constructed a phylogenetic tree of the strain and a large number of reported strains of the same genus, to identify the species. Finally, the strains of the same genus were put together to construct a phylogenetic tree by MEGA.
L130 “The samples”- what kind of samples, what the authors want to assess?
A: This mainly describes the methods of sample pretreatment for SEM.
L136 Analysis of the data was done . what kind of data? I suppose those of the bioassay. Move this section before SEM I
A:Yes, this data refer to bioassay data. I have modified this part.
Results
EPF species diversity in soils of Hebei and Central China
Why separate Hebei from other regions when the materials and methods have been reported as four regions. This section is quite confusing and cannot be reported as a list of species. All the results achieved should be introduced in the materials and methods so the groupings (regional/ crops vs grass vs follow lands) should be claimed before the aim of the study could be clear.
A: Because Hebei is not part of central China, it is appropriate to remove the central of the text (soils of China).
What result do you expect comparing fungi isolated from different regions? what do you expect comparing the outcomes from different soils…please explain
A: Comparing fungi isolated from different regions, we want to know which provinces are rich in soil fungi. Comparing the outcomes from different soils, we know that soil samples should be collected in places with rich vegetation, high humidity and suitable temperature, avoiding places with human activities.
302 pure fungal strains from 226 soil samples sound like a poor yield. I wonder if you performed a selection of the isolates. Please explain
A: Because these four provinces have a large proportion of agricultural land, humidity and temperature are not very suitable for fungal growth, and are also affected by the terrain
L190-192 Aspergillus, Fusarium, Lecanicillium, 189 Metarhizium, Monascus and Talaromyces all had obvious pathogenicityto P. striolata (Table 190 4). Besides, we found that Aspergillus, Lecanicillium, Monascus and Talaromyces had the pathogenicity. Genus and species cannot be used with the same meaning, especially when the general trait of a genus is not associated with the EP trait.
A: Multiple species in a genus are indicated by spp.
Discussion and conclusions are affected by the above mentioned weakness, so should be reworked.
A: OK, Discussion and Conclusions have reworked.
To sum up, we have carefully revised our manuscript. Please do not hesitate to contact us if you have any question. Thanks again for your hard work. Best wishes to you!
Author: Ke Zhang
Date: 14 May 2022
Reviewer 3 Report
The manuscript entitled "Entomopathogenic Fungi in soils of central China and their bioactivity against striped flea beetle Phyllotreta striolata" is well written and suitable to be considered for publication in this journal. However, some points are still required to be addressed before proceeding further. The points of revision are given below:
Comments:
- Spacing problem in line number 75 (beneficial to explore new), 79 (from sites), 84 (depth from three points was collected, mixed and), 190 (pathogenicity to).
- In line number 116: is the total number of individuals in one species add symbol stands for…
- Corrections:
Line number 8: Central China locate in the Yellow to Central China located in the Yellow…
Line number 9: and range from south subtropica to and ranged from south subtropica….
Line number 52: P. striolata (Coleoptera: Chrysomelidae) to Phyllotreta striolata (Coleoptera: Chrysomelidae)
Line number 62-63: that B. bassiana and M. anisopliae to that B. bassiana and M. anisopliae
Line number 63: and adults of P. striolata to and adults of P. striolata
Line number 72: this research is to to this research was to
Line number 74: environment changes on EPF are analysed to environment changes on EPF were analyzed
Line number 100: inoculated in PDA medium to inoculated on PDA medium
Line number 123: was prepared by culturing with light to were prepared by culturing with light
Line number 129: Scanning electron microscopic to Scanning electron microscopy
Line number 136: All data are expressed as to All data were expressed as
Line number 147: with 12, 2, 3, 1, 11 and 6 isolates found to with 12, 2, 3, 1, 11 and 6 isolates respectively
Line number 152: isolates were respectively identified as to isolates were identified as
Line number 161: Purpureocillium spp (A) and Metarhizium spp to Purpureocillium spp. (A) and Metarhizium spp.
Line number 161: Penicillium spp (A) and Aspergillus spp to Penicillium spp. (A) and Aspergillus spp.
Line number 161: tree of Talaromyces spp to tree of Talaromyces spp .
Line number 167: of Lecanicillium /Simplicillium spp to of Lecanicillium /Simplicillium spp.

Author Response
Dear Editor and Reviewer:
Thank you for taking time out of your busy schedule to review the manuscript.
I would like to thank you for suggestions on the manuscript. I have revised all possible errors highlighted in the article. Thank you again for your careful modification.
To sum up, we have carefully revised our manuscript. Please do not hesitate to contact us if you have any question. Thanks again for your hard work. Best wishes to you!
Author: Ke Zhang
Date: 14 May 2022
Round 2
Reviewer 1 Report
Accept for publication after checking journal format and text editing
Author Response
Dear Editor and Reviewer:
Thank you to review the manuscript again.
We have carefully revised our manuscript. Please do not hesitate to contact us if you have any question. Thanks again for your hard work. Best wishes to you!
Author: Ke Zhang
Date: 22 May 2022

Reviewer 2 Report
The authors gave a new version of the manuscript, but many of the changes performed are cosmetic.
The core of the comment has been skipped and several of the additional notes rebounded. So that the major concerns previously raised are still pending.
As previously the paper misses the right balance between the two aims of the work: diversity and applicative. The applicative one overshadows the diversity study that should be instead preponderant and reliable due to the Journal's aim. Other concerns raised were on methods (isolation and identification) and English both still pending. Indeed, the medium used is not selective for entomopathogenic fungi and molecular species identification is not reliable for all strains if based on a single marker.
Below are some additional notes
· it was asked to arrange the abstract following the common rules. It is still patchy and misses basic information (e.g. how you tackled the research question).
· Despite the links given to help authors in choosing keywords, only Isaria javanica can be considered a true kw, the others should be contextualized or discarded.
· Introduction. The question on Bacillus thuringensis has been raised to push the authors for more accurate background information but few changes have been performed. Background information is still missing or reported in a questionable order missing linearity. For example, the authors talk about resistance before introducing the concept; in addition, the search for sustainable pest controls is also due to the impact of pesticides on the environment – a missing item-. The authors should remember that what is obvious to the author may not be to the readers.
· Materials and methods. It was evidenced as the medium used is not a selective medium for entomopathogenic fungi, anyway, it is still claimed to be (false information). The temperature of incubation has been added but not the time of incubation. The PCR program used is still missing, and ITS (as raised before) is not enough for a reliable species identification within huge genera such as Aspergillus, Fusarium, Penicillium, Cladosporium, Trichoderma etc
· Results as before 300 fungal isolated are a poor result compared to the high number of samples taken. The reply given by the authors to the concern previously raised is vague and far to be complete because spores are everywhere. No changes have been applied to trees and the phylogeny cannot be considered reliable for the abovementioned concerns. About pathogenicity, it is not clear if the spore charge (10E4, 10E5, …10E8) is associated or not with the recorded mortality.
No further comments have been done on the Discussion and conclusions being a direct consequence of the previous flawed sections.
Some detailed comments are reported below
L20-21 please rework the sentence: it is confusing
L23 EPF, also known as entomogenous fungi,- discard because information already present at L19
L27The use of fungicides has brought great economic, ecological and social benefits [6]…and then? What’s the link with EPF?
L29 According to incomplete statistics, about.. what does it mean?
L73 were collected in different sites…This sounds obvious. You instead included different kind of soil. Please rework the sentence accordingly.
L75 The longitude and latitude in each site 75 were recorded by ICEGPS 100C (Shenzhen, China). That’s good but in the present paper there is no trace of them because you discarded the map.
L80-85 As before the medium used is not selective for EPF because rose /red Bengala slows down only. About the medium you can add the suppliers. PDA recipe is out of place and not necessary. Is instead missing the incubation time (e.g. 3 days, one week, 2 weeks, a month).
L88-89 please rework the sentence.
L93 and the standard PCR cycling protocol ..please report the protocol you used
L94 through the BLAST of NCBI. Last time I asked for constraint applied you didn’t reply to my question. So, I repeat, perhaps did you excluded environmental/uncultured sequences or limit your search to type strains?
L96 the citation 27 is not in the right place
L97 how did you choose the JC model? How many positions have been considered, how many strains for each dataset, outgroup, etc
L97 The standard EPF strains were referred (Table 1). What does it mean? In table 1 are reported strains not known to be entomopathogenic. So, it is important to know what the authors consider as “standard”. Most probably you mean reference strains; in this case type strains should be indicated and as note should be reported the meaning of the culture collection acronym complete of the city where they are stored.
Similarly, the title of table 1 should be changed
L107 As the reference [27]. As before the reader needs smart information not your citation. Please clarify how you selected the strains for bioassay.
L107-108 it is not clear why you prepared different spore suspensions. Did you expose insects to different spore solution?
Purpureocillium and Penicillium cannot be shortened both as P. it is confusing
Author Response
Dear Editor and Reviewer:
Thank you to review the manuscript again.
I would like to thank you for suggestions on the manuscript.
it was asked to arrange the abstract following the common rules. It is still patchy and misses basic information (e.g. how you tackled the research question).
Abstract: (1) Background: The present research was to explore the occurrence and diversity of entomopathogenic fungi (EPF) in cultivated and uncultivated land (crop, grass, fallowland and arbor) from provinces of China, and to search for EPF of Phyllotreta striolata; (2) Methods: Fungi were isolated from collected soil samples, identified and analyzed, and their activity against the adult of P. striolata were determined; (3) Results: 188 EPF isolates were identified from 226 soil samples, Hubei Province has best EPF diversity, Isaria javanica (IsjaHN3002) had the highest mortality, and the Aspergillus spp. was first reported about their entomopathogenic activities against P. striolata; (4) Conclusions: The amount and types of fungi in soil vary by region and vegetation, and soil is one of the resources for acquiring EPF.
- Despite the links given to help authors in choosing keywords, only Isaria javanica can be considered a true kw, the others should be contextualized or discarded.
A: Keywords: pathogenicity; species diversity; Isaria javanica
- Introduction. The question on Bacillus thuringensis has been raised to push the authors for more accurate background information but few changes have been performed. Background information is still missing or reported in a questionable order missing linearity. For example, the authors talk about resistance before introducing the concept; in addition, the search for sustainable pest controls is also due to the impact of pesticides on the environment – a missing item-. The authors should remember that what is obvious to the author may not be to the readers.
- Materials and methods. It was evidenced as the medium used is not a selective medium for entomopathogenic fungi, anyway, it is still claimed to be (false information). The temperature of incubation has been added but not the time of incubation. The PCR program used is still missing, and ITS (as raised before) is not enough for a reliable species identification within huge genera such as Aspergillus, Fusarium, Penicillium, Cladosporium, Trichoderma etc
- Results as before 300 fungal isolated are a poor result compared to the high number of samples taken. The reply given by the authors to the concern previously raised is vague and far to be complete because spores are everywhere. No changes have been applied to trees and the phylogeny cannot be considered reliable for the abovementioned concerns. About pathogenicity, it is not clear if the spore charge (10E4, 10E5, …10E8) is associated or not with the recorded mortality.
No further comments have been done on the Discussion and conclusions being a direct consequence of the previous flawed sections.
A: “ITS sequences are small and easy to analyze, have been widely used in phylogenetic analysis of different fungal species, but their accuracy is controversial, therefore, the identification of fungal species in this paper had some defects.”, “The isolation of EPF was not high, which showed that soil fungi were not abundant in these areas, and the sampling and isolation methods also affected the isolation of fungi.”added in Discussion
L20-21 please rework the sentence: it is confusing
A: In addition to absorbing nutrients for their own growth, some EPF can control insect populations at low levels for long periods.
L23 EPF, also known as entomogenous fungi,- discard because information already present at L19
A: OK, thank you for your advice.
L27The use of fungicides has brought great economic, ecological and social benefits [6]…and then? What’s the link with EPF?
A: OK, This sentence is abrupt, delete and modify.
L29 According to incomplete statistics, about.. what does it mean?
A: This is described in the reference literature. After all, worldwide surveys are difficult to cover, and only approximate data.
L73 were collected in different sites…This sounds obvious. You instead included different kind of soil. Please rework the sentence accordingly.
A: The soil samples were collected in different sites (cropland, fallowland, arbor and grass).
L75 The longitude and latitude in each site 75 were recorded by ICEGPS 100C (Shenzhen, China). That’s good but in the present paper there is no trace of them because you discarded the map.
A: The attached table (Table A1) shows the latitude and longitude of the sampling site.
L80-85 As before the medium used is not selective for EPF because rose /red Bengala slows down only. About the medium you can add the suppliers. PDA recipe is out of place and not necessary. Is instead missing the incubation time (e.g. 3 days, one week, 2 weeks, a month).
A: This medium is not selective for EPF, screening for fungi except yeast, cycloheximide can inhibit yeast, molds and protozoon. Because different fungi grow at different rates, we usually culture for 1 to 2 weeks.
L88-89 please rework the sentence.
A: The sentence was misplaced and corrected.
L93 and the standard PCR cycling protocol ..please report the protocol you used
A: the standard PCR cycling protocol ( 94 °C 3 min, 94 °C 30 s, 55 °C 30 s, 72 °C 1 min, 33 cycles; 72 °C 10 min).
L94 through the BLAST of NCBI. Last time I asked for constraint applied you didn’t reply to my question. So, I repeat, perhaps did you excluded environmental/uncultured sequences or limit your search to type strains?
A: Strains may show up in different species by BLAST, which we will confirm again by constructing phylogenetic tree.
L96 the citation 27 is not in the right place
A: OK, thank you for your advice.
L97 how did you choose the JC model? How many positions have been considered, how many strains for each dataset, outgroup, etc
A: This is just a model for building phylogenetic tree, and every model should make sense, but I don't know how the software works.
L97 The standard EPF strains were referred (Table 1). What does it mean? In table 1 are reported strains not known to be entomopathogenic. So, it is important to know what the authors consider as “standard”. Most probably you mean reference strains; in this case type strains should be indicated and as note should be reported the meaning of the culture collection acronym complete of the city where they are stored.
Similarly, the title of table 1 should be changed
A: This is a misstatement, which means that the reference strains used in the phylogenetic tree in this paper are in the table 1.
The phylogenetic trees of fungi were constructed by MEGA X with a statistical method of maximum likelihood, bootstrap test of 500 replications and the Jukes-Cantor model [27]. The standard fungi strains were referred (Table 1).
L107 As the reference [27]. As before the reader needs smart information not your citation. Please clarify how you selected the strains for bioassay.
A: In fact, we have carried out preliminary experiments on the all different strains, and 47 strains with good activity are listed in this paper.
L107-108 it is not clear why you prepared different spore suspensions. Did you expose insects to different spore solution?
A: The concentration of spore suspension used for virulence determination of 47 fungal strains was 1.0×108 spores/mL, added in the comments to Table 4.
Purpureocillium and Penicillium cannot be shortened both as P. it is confusing
A: OK, thank you for your advice.

This manuscript is a resubmission of an earlier submission. The following is a list of the peer review reports and author responses from that submission.
Round 1
Reviewer 1 Report
Please see attached my comments

Author Response
Dear Editor and Reviewer:
Thank you for taking time out of your busy schedule to review the manuscript.
I would like to thank you for suggestions on the manuscript and for reviewing the references. Based on the comments of reviewers, I have revised all possible errors highlighted in the article. Thank you again for your careful modification.
To sum up, we have carefully revised our manuscript. Please do not hesitate to contact us if you have any question. Thanks again for your hard work. Best wishes to you!
Author: Ke Zhang
Date: 15 Mar 2022

Reviewer 2 Report
This paper is useful for researchers in the field. However, mode of mechanism against Phyllotreta striolata is not detected in the paper. Some mechanisms with chemicals relating to the mode of action are needed to add in the paper.
Author Response
Dear Editor and Reviewer:
Thank you for taking time out of your busy schedule to review the manuscript.
I would like to thank you for suggestions about the mode of mechanism against Phyllotreta striolata, our team will delve into the mechanism of action.
Thanks again for your hard work. Best wishes to you!
Author: Ke Zhang
Date: 15 Mar 2022

Reviewer 3 Report
In the submitted manuscript by Zhang et al., entitled “Entomopathogenic Fungi in Soils of Central China and the bioactivity against Phyllotreta striolata” the author isolate entomopathogenic fungi from soil collected in Central China. The obtained fungi were identified based on morphology and ITS sequences. Their ability to against the flea beetle was evaluated.
Overall, the experiments included in this manuscript are informative and interpretations seem reasonable. The manuscript writing is understandable and fairly easy to read. However, as I detail below, a few issues should be addressed:
- Title should be change to “Entomopathogenic Fungi in Soils of Central China and their ability to against the flea beetle, Phyllotreta striolata”
- The abstract should reflect the problem, the method, and the conclusions. It should be improved further.
- Introduction is not sufficient. Please add more related papers; clearly state the importance of the topic, recent examples from the literature. It would be informative to include the overall loss in: why do authors focus on against flea beetle? How dose flea beetle affect in Chinese agriculture?
- The procedure of Scanning electron microscopic (SEM) observation of infection process of selected entomopathogenic Fungi should be described in Materials & Methods section.
- Results should be improved. Fungal name in Table 1 should be italic. Phylogenetic tree should be improved with high resolution. Type species should be indicated in different from the obtained fungal isolates. Why does the author choose only 47 fungal isolates for pathogenicity test of flea beetle?
- The pathogenicity tests conducted using selected fungal isolates in the experiment have confirmed their pathogenicity in a flea beetle (Table 4). However, statistical analysis of these experiments must be performed using an appropriate statistical method to know whether the results are statistically significant or not. Figure 8A should be changed.
- Discussion looks more like introduction. It must be modified in such a way that it should include the experiments performed and the significant findings deduced that aided in confirming the main objective of the study.
- Conclusion should be added that include major outcome of your study in a succinct manner.
- I also found many typos errors, therefore it requires native speaker to proofread on it. Some suggestions are in the attached file.

Author Response
Dear Editor and Reviewer:
Thank you for taking time out of your busy schedule to review the manuscript. Now we have carefully correted and replied the manuscript for this revision. The revision instructions are as follows:
Q1: Title should be change to “Entomopathogenic Fungi in Soils of Central China and their bioability against striped flea beetle, Phyllotreta striolata”
A: The title has been changed to “Entomopathogenic Fungi in Soils of Central China and their ability to against the flea beetle, Phyllotreta striolata”.
Q2: The abstract should reflect the problem, the method, and the conclusions. It should be improved further.
A:The abstract modified as “Four provinces, Hunan, Hubei, Henan and Hebei in Central China locate in the Yellow and Yangtze river basins and range from south subtropical to north temperate zone with the diverse landforms and climate types, nevertheless the distribution of entomopathogenic fungi (EPF) in these areas is inadequately understand. The present research was to explore the occurrence and diversity of EPF in cultivated and uncultivated land (crop, grass, fallowland and arbor) from above four provinces. The striped flea beetle Phyllotreta striolata (Coleoptera : Chrysomelidae) is a major pest of cruciferous vegetables in southern China, in order to search for EPF of P. striolata, we determined the activity of 47 fungal strains agaist the adult of P. striolata. The results indicated that Isaria javanica (IsjaHN3002) had the highest mortality, and the Aspergillus were first reported about their entomopathogenic activities against P. striolata. Our experiment will give new insights to understanding of EPF distribution characteristics and their biodiversities conservation.”
Q3: Introduction is not sufficient. Please add more related papers; clearly state the importance of the topic, recent examples from the literature. It would be informative to include the overall loss in: why do authors focus on against flea beetle? How dose flea beetle affect in Chinese agriculture?
A:This part has been rewritten in accordance with the suggestions.
Q4: The procedure of Scanning electron microscopic (SEM) observation of infection process of selected entomopathogenic Fungi should be described in Materials & Methods section.
A: Thanks for your advice. The SEM method is supplemented, as follow:
2.6 Scanning electron microscopic (SEM)
The samples were placed in a 2 mL centrifuge tube, fixed with 2.5% glutaraldehyde overnight, washed with physiological saline and dehydrated using a graded series of ethanol, isoamyl acetate was replaced overnight. They were vacuum-dried, fixed onto the platform and then coated with platinum with an ion coater before being observed using a scanning electron microscope.
Q5: Results should be improved. Fungal name in Table 1 should be italic. Phylogenetic tree should be improved with high resolution. Type species should be indicated in different from the obtained fungal isolates. Why does the author choose only 47 fungal isolates for pathogenicity test of flea beetle?
A: Results and discussion Fungal name in Table 1 have changed to italic. Phylogenetic trees also have changed to higher resolution images. We mainly from the morphological selection of different strains for activity.
Q6: The pathogenicity tests conducted using selected fungal isolates in the experiment have confirmed their pathogenicity in a flea beetle (Table 4). However, statistical analysis of these experiments must be performed using an appropriate statistical method to know whether the results are statistically significant or not. Figure 8A should be changed.
A: Duncan's new complex range test was used for statistical analysis of the data, and the significant differences could not be expressed in all letters, so there is no significant difference marked, only the standard deviation is added. All images have been modified as shown in Figure 8 below.
Q7: Discussion looks more like introduction. It must be modified in such a way that it should include the experiments performed and the significant findings deduced that aided in confirming the main objective of the study. Conclusion should be added that include major outcome of your study in a succinct manner.
A: Discussion and conclusion are reorganized.
Q8: I also found many typos errors, therefore it requires native speaker to proofread on it. Some suggestions are in the attached file.
A:We have asked the English major teacher to modify the grammar of the whole paper, combining all the suggestions in the attachment.
To sum up, we have carefully revised our manuscript. Please do not hesitate to contact us if you have any question. Thanks again for your hard work. Best wishes to you!
Author: Ke Zhang
Date: 15 Mar 2022

Reviewer 4 Report
Review Diversity, Zhang et al.
I appreciated the opportunity to review this manuscript titled “Entomopathogenic Fungi in Soils of Central China and the bioactivity against Phyllotreta striolata”, which with some work, it will be a valuable contribution to the knowledge of entomopathogenic fungi (EPF) biodiversity and how the selection of native strains could help to control specific local pests. However, the manuscript presents important issues that need to be strongly considered and improved. One of the most important problems I found through the paper have been the wording and the use of English, which is poor and far to be the appropriate for a scientific publication, to the point of being completely understandable in some parts. In addition, the terminology and concepts described for the study of EPF biodiversity and pathogenicity (better than bioactivity) are not the appropriate. I strongly recommend studying other authors that are great experts on the study of EPF diversity and pathogenicity (Meyling, Eilenberg, Enkerli, Quesada-Moraga, etc.), some of them already cited in your manuscript. In addition, the introduction is very general and the information very imprecise. Please, try to be more concise and focus the introduction on the topic of the paper: EPF biodiversity in soils and the selection of native strains to control of Phyllotreta striolata. In fact, you didn’t mention anything about Phyllotreta striolata through the introduction. Why did you choose this species? What do you know about it? Why are you looking alternatives to replace chemical compounds for its control? etc. On the other hand, the methodology is a bit chaotic, it is poorly described, and impossible to reproduce it. For example, the bioassays are very superficially described. Even if you referred the methodology followed from other paper, you should provide more information. That is, how many repetitions did you performed, insects per repetition, diagnosis of the death, etc. You should demonstrate Koch postulates when you describe a new strain that could be a new biological control agent. Regarding to the molecular analysis, in my opinion, the resolution of ITS1-ITS4 is not the best to study EPF diversity. In the literature, there are many other genes (EF1a, Bloc, etc.), that could be analyzed separately or in combination between them and with other technics (SSRs, SNPs, etc.), which have shown great results to study EPF biodiversity (see the authors mentioned above). Moreover, I don’t really understand why the authors have included many other fungi that have not been described as EPF. You should focus on EPF, which will simplify a lot the experiments, costs, analysis, and interpretation. Same for the sequencing analysis and biodiversity statistics. You have many other ways to analysis alpha and beta diversity of a population and/or community. The biodiversity analysis performed in your manuscript is very weak and superficial. To continue, results are difficult to follow, figures are complicate to interpret, tables and figures need to be self-explanatory, and it is not the case in most of the cases, the map is very confusing, etc.. I could continue with a long list of things that should be improved, but I have mentioned just the most critical ones. As soon as you resolve these issues, the paper will be in a better stage to be published in a journal such as Diversity. Unfortunately, due to all the comments mentioned above, I must recommend rejecting the manuscript.
Author Response
Dear Editor and Reviewer:
Thank you for taking time out of your busy schedule to review the manuscript. Now we have carefully correted and replied the manuscript for this revision. The explanation is as follows:
Q:One of the most important problems I found through the paper have been the wording and the use of English, which is poor and far to be the appropriate for a scientific publication, to the point of being completely understandable in some parts.
A:We have asked the English major teacher to revise the whole text in terms of grammar and expression.
Q:the terminology and concepts described for the study of EPF biodiversity and pathogenicity (better than bioactivity) are not the appropriate. I strongly recommend studying other authors that are great experts on the study of EPF diversity and pathogenicity (Meyling, Eilenberg, Enkerli, Quesada-Moraga, etc.), some of them already cited in your manuscript.
A:We changed the terminology and concepts described for the study of EPF biodiversity and pathogenicity, thank for your advice.
Q:The introduction is very general and the information very imprecise. Please, try to be more concise and focus the introduction on the topic of the paper: EPF biodiversity in soils and the selection of native strains to control of Phyllotreta striolata. In fact, you didn’t mention anything about Phyllotreta striolata through the introduction. Why did you choose this species? What do you know about it? Why are you looking alternatives to replace chemical compounds for its control? etc.
A:The introduction part was really badly written. I rewrote it according to the comments of reviewers. The introduction as follows:
Q: the methodology is a bit chaotic, it is poorly described, and impossible to reproduce it. For example, the bioassays are very superficially described. Even if you referred the methodology followed from other paper, you should provide more information. That is, how many repetitions did you performed, insects per repetition, diagnosis of the death, etc. You should demonstrate Koch postulates when you describe a new strain that could be a new biological control agent. Regarding to the molecular analysis, in my opinion, the resolution of ITS1-ITS4 is not the best to study EPF diversity. In the literature, there are many other genes (EF1a, Bloc, etc.), that could be analyzed separately or in combination between them and with other technics (SSRs, SNPs, etc.), which have shown great results to study EPF biodiversity (see the authors mentioned above).
A:The method was described in detail, We also recognize the vulnerability of molecular analysis, More strict strain identification and in-depth research will be carried out for the screened strains with high virulence.
Q: Moreover, I don’t really understand why the authors have included many other fungi that have not been described as EPF. You should focus on EPF, which will simplify a lot the experiments, costs, analysis, and interpretation. Same for the sequencing analysis and biodiversity statistics. You have many other ways to analysis alpha and beta diversity of a population and/or community. The biodiversity analysis performed in your manuscript is very weak and superficial.
A: Our aim was to find highly virulent strains and to contribute new strains to the diversity of EPF,new EPF may be found in strains that have never been described as EPF.
Q: results are difficult to follow, figures are complicate to interpret, tables and figures need to be self-explanatory, and it is not the case in most of the cases, the map is very confusing, etc..
A:The results,tables and figures have been revised. The map shows where we collected the soil samples, and the attached table details the specific sampling sites.
To sum up, we have carefully revised our manuscript. Please do not hesitate to contact us if you have any question. Thanks again for your hard work. Best wishes to you!
Author: Ke Zhang
Date: 15 Mar 2022

Reviewer 5 Report
These are my main comments on the MS (diversity-1616087) entitled:“Entomopathogenic Fungi in Soils of Central China and the bioactivity against Phyllotreta striolata”
Generally, it is an interesting study investigating the presence of various EPF in China regions and their action against an insect pest.
Apart from poor language (grammatical and syntax errors), I have located many serious flaws in the MS. Authors should make a lot of corrections before publication (I have suggested many changes in the PDF). My main objections focused on the methodology of the bioassay, result presentation and most of all the discussion which should be re-written.
My proposal is to reject it for publication in "DIVERSITY". Authors may follow my suggestions on the attached PDF to improve their MS before re-submission.

Author Response
Dear Editor and Reviewer:
Thank you for taking time out of your busy schedule to review the manuscript. Now we have carefully correted and replied the manuscript for this revision. The explanation is as follows:
Q:Apart from poor language (grammatical and syntax errors), I have located many serious flaws in the MS. Authors should make a lot of corrections before publication (I have suggested many changes in the PDF).
A:We have asked the English major teacher to modify the grammar and syntax of the whole paper, combining all the suggestions in the attachment.
Q:My main objections focused on the methodology of the bioassay, result presentation and most of all the discussion which should be re-written.
A:The methodology , result, discussion and conclusion have been re-written. The discussion and conclusion as follows:
To sum up, we have carefully revised our manuscript. Please do not hesitate to contact us if you have any question. Thanks again for your hard work. Best wishes to you!
Author: Ke Zhang
Date: 15 Mar 2022

Round 2
Reviewer 3 Report
Dear Author,
Thank you very much for improving your manuscript. This version suitable for publish in this Diversity-MDPI.
Reviewer 4 Report
2nd Review Diversity, Zhang et al.,
Dear Mr. Zhang et al.,
I appreciate the big effort you made to improve your manuscript based on all our comments and the answers provided. With this, the manuscript has reached a new level. However, some of the main concern I found critical to reject the manuscript in the first revision step, they have not been resolved, for example:
1.- Even after the English syntax/grammar improvement, I still feel that the document does not flow properly. It is difficult to follow some parts of the document, there are many points where the information is not precise or incorrect.
2.-Regarding to the comment in the first round of the review “I strongly recommend studying other authors that are great experts on the study of EPF diversity and pathogenicity (Meyling, Eilenberg, Enkerli, Quesada-Moraga, etc.)”, I am missing many of the authors I suggested you through the introduction and the discussion.
3.-The methodology, among other issues, I still miss one of the most important points: the insect death diagnosis. In addition, I am also missing the control results, which do not appear anywhere though the manuscript.
4.- Authors have not included many entomopathogenic fungi in the alignments, for example Beauveria and Metarhizium. It has been described around 30 species of Metarhizium, and you only include 4 in your alignment… (See the paper of Bischoff, J.F.; Rehner, S.A.; Humber, R.A. A multilocus phylogeny of the Metarhizium anisopliae lineage. Mycologia. 2009, 101, 512–530, it will help you). However, the authors are providing the alignments of isolated, but in my opinion, the figures are heavy, the clustering is not very good (to resolve it you should include more sequences from databases), very small, with not too many information, and in your shoes, I will include the alignment in supplementary information, and give more analysis about diversity. You could play a lot with your data, the distribution, the importance of the land management, etc. As I mentioned in my previous revision, you should improve the diversity analysis if the word “biodiversity” is in the keywords.
I will try to give you some more comment in the attached PDF.
Due to all this reasons, unfortunately I have considered that your manuscript is not still ready for publication.

Reviewer 5 Report
The MS has been changed, but most of my main comments have been ignored. Even some typos have not been corrected. Authors should have provided a point by point response to my comments. Results and Discussion still need to be improved. Especially the Discussion. The bioassay is not discussed at all. Although the MS includes many useful data it cannot be published in this form.